# Emergency triage of brain computed tomography via anomaly detection with a deep generative model

Seungjun Lee [1,5], Boryeong Jeong [2,5], Minjee Kim [1], Ryoungwoo Jang[1], Wooyul Paik [3], Jiseon Kang [2], Won Jung Chung[4], Gil-Sun Hong [2,6✉] & Namkug Kim [1,2,6✉]

Triage is essential for the early diagnosis and reporting of neurologic emergencies. Herein, we report the development of an anomaly detection algorithm (ADA) with a deep generative model trained on brain computed tomography (CT) images of healthy individuals that reprioritizes radiology worklists and provides lesion attention maps for brain CT images with critical findings. In the internal and external validation datasets, the ADA achieved area under the curve values (95% confidence interval) of 0.85 (0.81–0.89) and 0.87 (0.85–0.89), respectively, for detecting emergency cases. In a clinical simulation test of an emergency cohort, the median wait time was significantly shorter post-ADA triage than pre-ADA triage by 294 s (422.5 s [interquartile range, IQR 299] to 70.5 s [IQR 168]), and the median radiology report turnaround time was significantly faster post-ADA triage than pre-ADA triage by 297.5 s (445.0 s [IQR 298] to 88.5 s [IQR 179]) (all $p < 0.001$).

[1] Department of Convergence Medicine, University of Ulsan College of Medicine, Asan Medical Center, Seoul, Republic of Korea. [2] Department of Radiology and Research Institute of Radiology, University of Ulsan College of Medicine, Asan Medical Center, Seoul, Republic of Korea. [3] Department of Radiology, Gangneung Asan Hospital, University of Ulsan College of Medicine, Gangneung, Republic of Korea. [4] Department of Health Screening and Promotion Center, University of Ulsan College of Medicine, Asan Medical Center, Seoul, Republic of Korea. [5] These authors contributed equally: Seungjun Lee, Boryeong Jeong. [6] These authors jointly supervised this work: Gil-Sun Hong, Namkug Kim. ✉email: hgs2013@gmail.com; namkugkim@gmail.com

Neurological emergencies should be diagnosed and treated as soon as possible to reduce mortality and morbidity rates and to enhance functional outcomes[1–3]. For the initial screening and diagnosis of neurological conditions, non-contrast brain computed tomography (CT) is the current standard imaging modality. In this regard, radiology worklist reprioritization based on image findings is critical in the emergency department (ED).

With the excellent achievements of deep learning in various radiological tasks, several studies have demonstrated that deep learning-based radiological triage can improve radiology workflow efficiency, accelerate radiology reporting, and enable timely management of patients with critical findings (e.g., intracranial hemorrhage or large vessel occlusion on brain images)[4–7]. However, data-related problems have restricted the broad clinical application of deep learning. The construction of large-scale annotated training datasets across diverse populations, disease entities from common to rare, medical centers, and acquisition protocols has remained a significant obstacle to developing a deep learning system in medicine. In addition, the clinical efficacy of supervised deep learning models has been validated only in selected patients with the risk of having a single disease or a few specific diseases. Therefore, this approach cannot guarantee that deep learning can cope with new or previously unseen conditions. As a result, the clinical applicability of supervised deep learning with a narrow clinical focus has been limited.

Recently, pilot studies have shown that deep generative models trained on normal data can detect anomalies[8–13]. Deep generative models learn to capture target data distribution; hence, they can detect anomalous data that deviate from the target distribution without prior knowledge of anomalies. Moreover, the anomaly detection framework based on deep generative models can visually highlight the model's prediction using reconstruction error. Although previous studies using this framework have attracted considerable attention, they have two limitations: (1) lack of external and clinical validation tests (hence, whether the model can be generalized to real-world situations cannot be guaranteed) and (2) no clinical utility test of them.

This study aimed to develop and validate an anomaly detection algorithm (ADA) based on a deep generative model trained only with normal brain CT images and investigate the clinical impact of an ADA-based triage system on ED radiology workflow using a randomized crossover clinical simulation test. Importantly, this study aimed to assess the real-world performance of the ADA using brain CT images in internal and external ED screening cohorts.

## Results

We developed an ADA based on a deep generative model called the closest normal-style-based generative adversarial network (CN-StyleGAN). Using brain CT images from healthy individuals, CN-StyleGAN was trained to reconstruct a scan into the closest normal-style scan. The density error between the actual scan and the reconstructed scan was used to determine the anomaly score of the scan to identify emergency cases. Cases identified as emergency cases were reprioritized based on their anomaly scores in the radiology worklist as well as the visualization of the predicted lesions (Fig. 1).

**Baseline characteristics of the training, tuning, and validation datasets**. Figure 2 and Supplementary Table 1 summarize the data collection, baseline characteristics, and image acquisition information for the datasets. Non-contrast brain CT scans from 34,085 healthy individuals (mean age ± standard deviation [SD]: 42.9 ± 19.6 years; female: 18,232 [53.5%]) were retrospectively collected from a tertiary academic hospital for the training dataset. Furthermore, brain CT scans were collected independently and retrospectively from consecutive individuals who underwent emergency screening for suspected neurological conditions in the EDs of an internal and an external institution. The internal dataset included brain CT scans from 544 individuals (mean age ± SD: 58.6 ± 17.8 years; female: 280 [51.5%]) who had visited the ED of the internal institution for one month. Following that, the internal dataset was randomly divided into two parts: the tuning dataset and the internal validation dataset. The external validation dataset included brain CT scans from 1795 consecutive individuals (mean age ± SD: 60.3 ± 19.3 years; female: 875 [48.7%]) who had visited the ED of an external institution for five months. For the tuning and internal and external validation datasets, each case was classified into one of the five emergency categories: normal, benign, indeterminate, urgent, and immediate (Supplementary Table 2). Subsequently, both urgent and immediate cases were defined as emergency cases that required emergency intervention, regardless of the neurological entity. The emergency cases accounted for 15.0% (41 of 273) and 11.0% (197 of 1795) of the internal and external validation datasets, respectively. Disease entities from the internal and external validation datasets included brain mass-like lesions (39.0% [16 of 41] vs. 10.2% [20 of 197]), acute infarctions (7.3% [3 of 41] vs. 19.8% [39 of 197]), intracranial hemorrhage (43.9% [18 of 41] vs. 65.0% [128 of 197]), hydrocephalus (4.9% [2 of 41] vs. 3.0% [6 of 197]), and other diseases (4.9% [2 of 41] vs. 2.0% [4 of 197]).

**Emergency case detection performance of the ADA**. The mean ± SD of the anomaly score was significantly different between the non-emergency and emergency groups in the internal and external validation tests (14.8 ± 36.9 vs. 98.6 ± 119.7, $p < 0.001$, and 14.5 ± 47.3 vs. 118.5 ± 177.3, $p < 0.001$, respectively) (Fig. 3a). The emergency case detection performance of the ADA was analyzed by calculating the area under the receiver operating characteristic (ROC) curve (AUC), sensitivity, specificity, and accuracy with 95% confidence intervals (CIs). The maximum value of Youden's index for the ROC curve analysis using the tuning dataset revealed the optimal anomaly score cutoff value. In the internal and external validation datasets, no data were excluded to reflect real data without sampling bias. Consequently, the AUC, sensitivity, specificity, and accuracy with 95% CIs were 0.85 (0.81–0.89), 0.71 (0.60–0.82), 0.78 (0.74–0.82), and 0.77 (0.73–0.80), respectively, in the internal validation test and 0.87 (0.85–0.89), 0.78 (0.74–0.82), 0.81 (0.80–0.83), and 0.81 (0.80–0.82), respectively, in the external validation test (Fig. 3b, Supplementary Fig. 1, and Supplementary Table 3). The false-negative rates were 29.3% (12 of 41 in the internal validation dataset) and 22.3% (44 of 197 in the external validation dataset). The false-positive rates were 22.4% (52 of 232 in the internal validation dataset) and 19.1% (305 of 1598 in the external validation dataset). For the detection of immediate cases, the ADA achieved the AUC values of 0.96 (0.94–0.99) and 0.95 (0.93–0.96) in the internal and external validation tests, respectively. According to disease entity, the AUC values with 95% CIs in the internal and external validation tests were as follows: brain mass-like lesions, 0.92 (0.88–0.96) vs. 0.92 (0.88–0.96); acute infarctions, 0.91 (0.86–0.95) vs. 0.87 (0.83–0.91); intracranial hemorrhages, 0.78 (0.70–0.85) vs. 0.86 (0.83–0.88); hydrocephalus, 0.82 (0.64–0.97) vs. 0.94 (0.92–0.97); and other diseases, 0.95 (0.91–0.99) vs. 0.80 (0.65–0.94) (Supplementary Fig. 2). Figure 4 shows representative cases of various diseases detected as emergency cases by the ADA and lesion attention maps provided by the ADA (see Supplementary Fig. 3 for representative false-positive and false-negative cases).

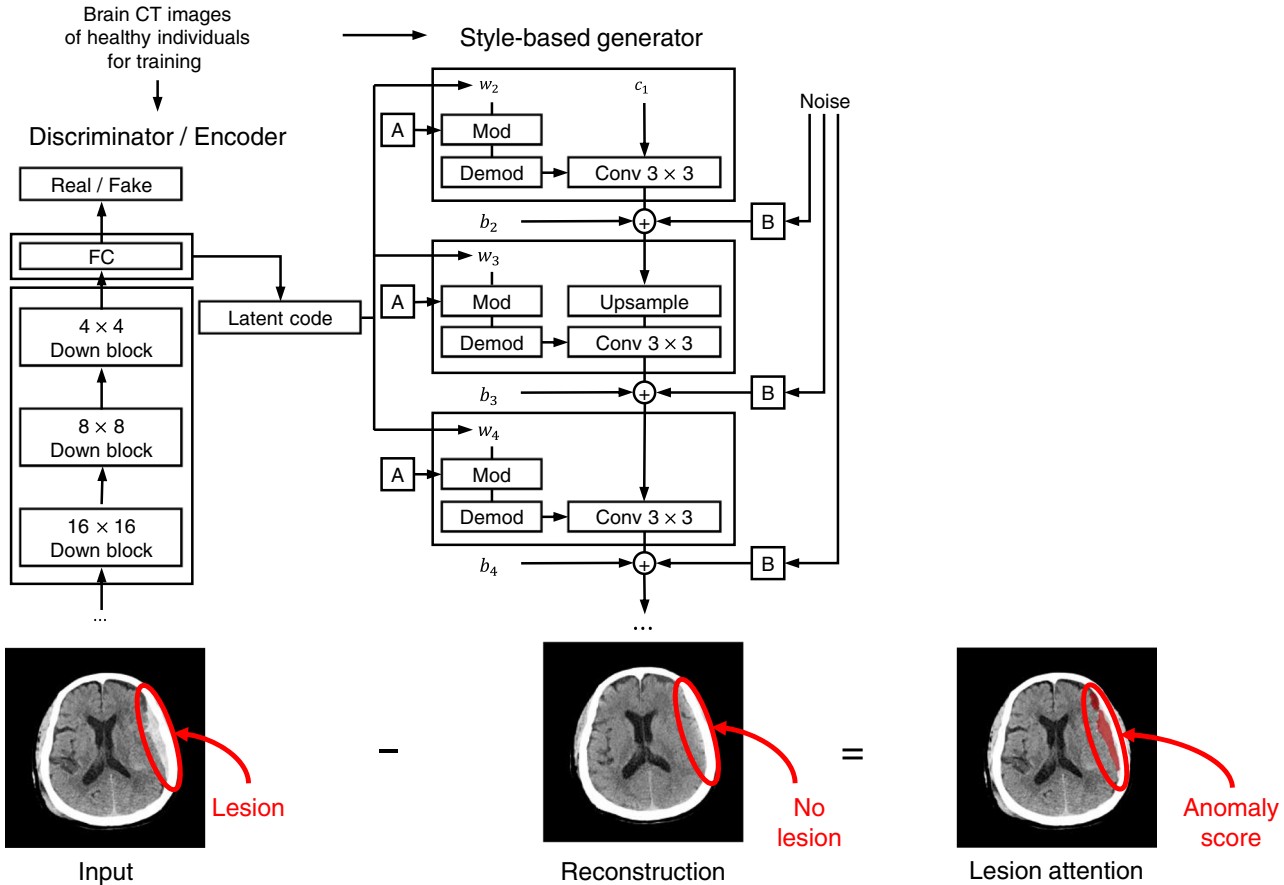

**Fig. 1 Our proposed anomaly detection framework based on a deep generative model— closest normal-style-based generative adversarial network (CN-StyleGAN)—trained on normal brain CT images.** The model reconstructs a brain CT image as its closest normal-style brain CT image. Based on the density error, the anomaly score of the scan is determined and used to identify emergency cases, followed by visualization of the detected lesion upon overlay.

Furthermore, sensitivities and specificities (95% CIs) were calculated, with the thresholds derived using the tuning dataset at high sensitivity levels of 0.95 and 1.00. At a sensitivity level of 0.95 for the tuning dataset, the sensitivity and specificity were 0.90 (0.83–0.97) and 0.60 (0.56–0.65), respectively, in the internal validation dataset and 0.89 (0.86–0.92) and 0.63 (0.62–0.65), respectively, in the external validation dataset. At a sensitivity level of 1.00 for the tuning dataset, the sensitivity and specificity were 1.00 (1.00–1.00) and 0.42 (0.37–0.47), respectively, in the internal validation dataset and 0.96 (0.95–0.98) and 0.47 (0.45–0.49), respectively, in the external validation dataset.

**Clinical simulation test for emergency case prioritization.** To investigate the clinical efficacy of the ADA-based triage system for radiology workflow, a randomized crossover study was performed in two sessions using the external validation dataset by referring to the existing study[14] (Fig. 5a). Two radiology experts on brain CT images independently participated in reviewing the images and reporting the critical findings using an in-house web-based user interface. A total of 1795 brain CT scans from the external validation dataset were randomized to two groups (groups A [898 brain CT scans] and B [897 brain CT scans]). In each group, brain CT scans were randomly assigned to 39 blocks (23 CT scans per block in group A, except for the last block, which included 24 CT scans; and 23 CT scans per block in group B). One block indicates the workload of a radiologist or emergency physician at one time. In the first reading session, each

reader assessed group A without the ADA and assessed group B with the ADA. In the second reading session, each reader assessed group A with the ADA and assessed group B without the ADA. The first and second sessions were separated by at least two weeks, and the reading orders of blocks were randomized for each reading session. The ADA-based triage system reprioritized emergency cases in the radiology worklists in each block and labeled them in red to alert readers.

The clinical efficacy of the ADA was analyzed according to three radiological time metrics based on previous studies[6,15,16]: wait time (WT; the time required to open a CT for image review from the beginning of one block), radiology report turnaround time (TAT; the time required to report a critical CT finding from the beginning of one block), and reading time (RT; the time between opening and closing a CT) for each case in each block.

Table 1 summarizes the outcomes before and after ADA implementation and presents them as median values in seconds (interquartile range [IQR]). In the emergency group, the median WT was significantly shorter post-ADA triage by 294 s (70.5 s [IQR 168]) than pre-ADA triage (422.5 s [IQR 299]) ($p < 0.001$). The median TAT was significantly faster post-ADA triage by 297.5 s (88.5 s [IQR 179]) than pre-ADA triage (445.0 s [IQR 298]) ($p < 0.001$). There was no significant difference in RT between pre-ADA and post-ADA triage (29.0 s [IQR 12.5] vs. 30.0 s [IQR 11.0], $p = 0.38$). As expected, in the non-emergency group, there was a significant delay in the WT and TAT when the ADA was implemented. However, the absolute difference in the WT and TAT between pre-ADA and post-ADA triage was

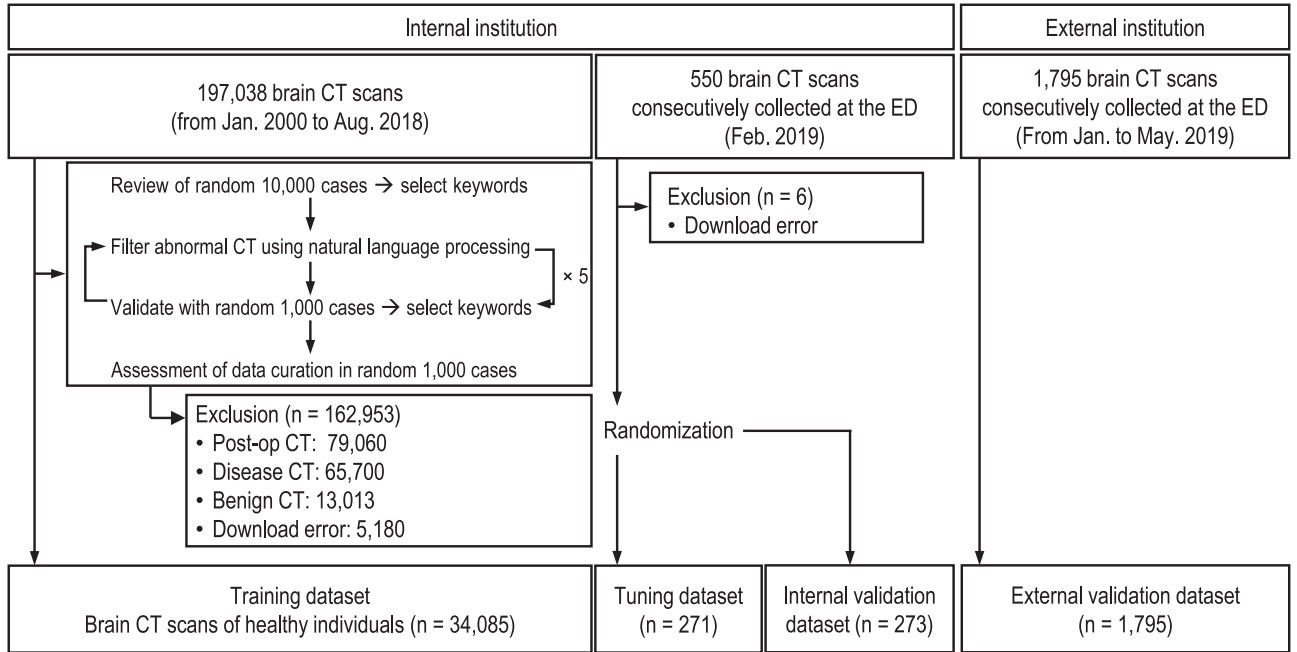

**Fig. 2 Data flow diagram of the collection and curation process of the training, tuning, internal validation, and external validation datasets.** The training dataset was collected and curated to include brain CT scans from healthy individuals by reviewing and applying NLP algorithms to radiological reports. In addition, consecutive brain CT scans from individuals who underwent emergency screening for suspected neurological conditions in the EDs of the internal and external institutions were independently and retrospectively collected. The internal dataset was randomly divided into two parts: a tuning dataset and an internal validation dataset. The external validation dataset included brain CT scans from 1795 consecutive individuals who had visited the ED of the external institution for five months.

significantly smaller in the non-emergency group (79.3 s [IQR 197.9] and 72.8 s [IQR 202.3]) than in the emergency group (294.0 s [IQR 352] and 297.5 s [IQR 347]) ($p < 0.001$). The RT was significantly shorter post-ADA triage by 1.5 s (31.00 s [11.5]) than pre-ADA triage (28.00 [11.5]) ($p < 0.001$). In the false negatives, the median WT and TAT were significantly delayed by 71 s and 70.3 s, respectively, post-ADA triage compared with pre-ADA triage (358.0 [IQR 291.5] to 449.8 s [IQR 199.3], $p = 0.009$ and 471.0 s [IQR 205] to 384.3 [IQR 300.9], respectively; $p = 0.02$) (Table 2). Figure 5b shows the significant reduction in the WT and TAT in the subgroups of emergency cases. Note that the WT and TAT were significantly shorter in the immediate group (350 s [260.3] and 355 s [266.6], respectively) than in the urgent group (245.5 s [422.5] and 245.5 s [439.5], respectively) (all $p = 0.002$).

**Discussion**

Our study proposed an anomaly detection approach based on a deep generative model trained only with normal brain CT images from healthy individuals. Although the proposed model did not reach the level of the supervised learning-based model performance, our study showed that the ADA has a clear advantage in terms of covering a diversity of diseases seen in the ED. In particular, our research demonstrated the potential clinical applicability of the ADA as a triage system for patients with emergency conditions.

Our research demonstrated the moderate but consistent performance of the ADA based on a deep generative model for internal and external validation datasets. Our external validation dataset represents real-world data that were consecutively collected from ED patients with neurologic symptoms and acquired from diverse CT machines and scanning protocols. Our results are supported by the findings of previous related studies in terms of the acceptable performance by an anomaly detection model

and good generalizability. Han et al.[17] reported on a GAN-based anomaly detection model with an AUC of 0.727–0.894 for detecting Alzheimer's disease and an AUC of 0.921 for detecting brain metastases from MRI. Choi et al.[9] reported on a deep learning model trained only using normal brain images to identify brain abnormalities (AUC of 0.74) in on brain positron emission tomography-CT (PET-CT) images. Fujioka et al.[10] proposed a GAN-based anomaly detection model with an AUC of 0.936 for distinguishing normal tissue from benign and malignant masses based on breast ultrasound imaging. These prior studies are valuable in that they demonstrated the capability of anomaly detection models in various medical images. However, the previous studies lacked external clinical validation tests; thus, whether these models can be generalized to real-world situations cannot be guaranteed. Therefore, further evidence with real-world data is warranted. Our study serves this purpose.

The other critical point of our study is that our research demonstrated the feasibility of our ADA as a triage system for brain CT scans in the ED. Our study revealed that ADA implementation significantly reduced the WT and TAT in emergency cases. Our results are comparable to those of previous studies regarding the clinical feasibility of patient triage by supervised anomaly detection models. Titano et al.[18] reported that their supervised model potentially raised the alarm 150 times faster than humans for urgent cases in brain CT scans. Wood et al.[19] demonstrated that the supervised anomaly detection model significantly reduced the mean reporting time for abnormal MRI examinations from 28 days to 14 days and from 9 days to 5 days for two hospital networks. Notably, in the detailed subgroup analysis of our study, ADA implementation led to a significant reduction in the WT and TAT in immediate (more urgent) cases than in urgent cases. This is because ADA-based classification is based on anomaly scores. A higher anomaly score for a limited intracranial space likely reflects a correspondent urgency on an

a

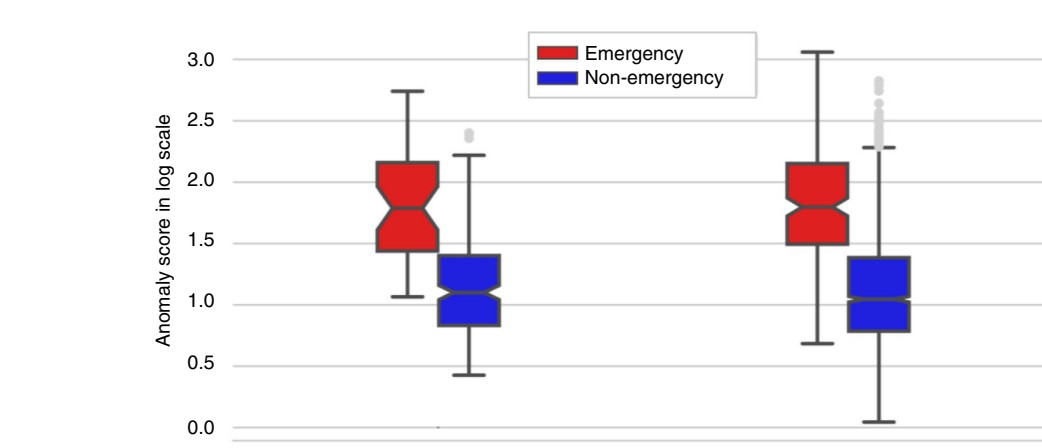

b

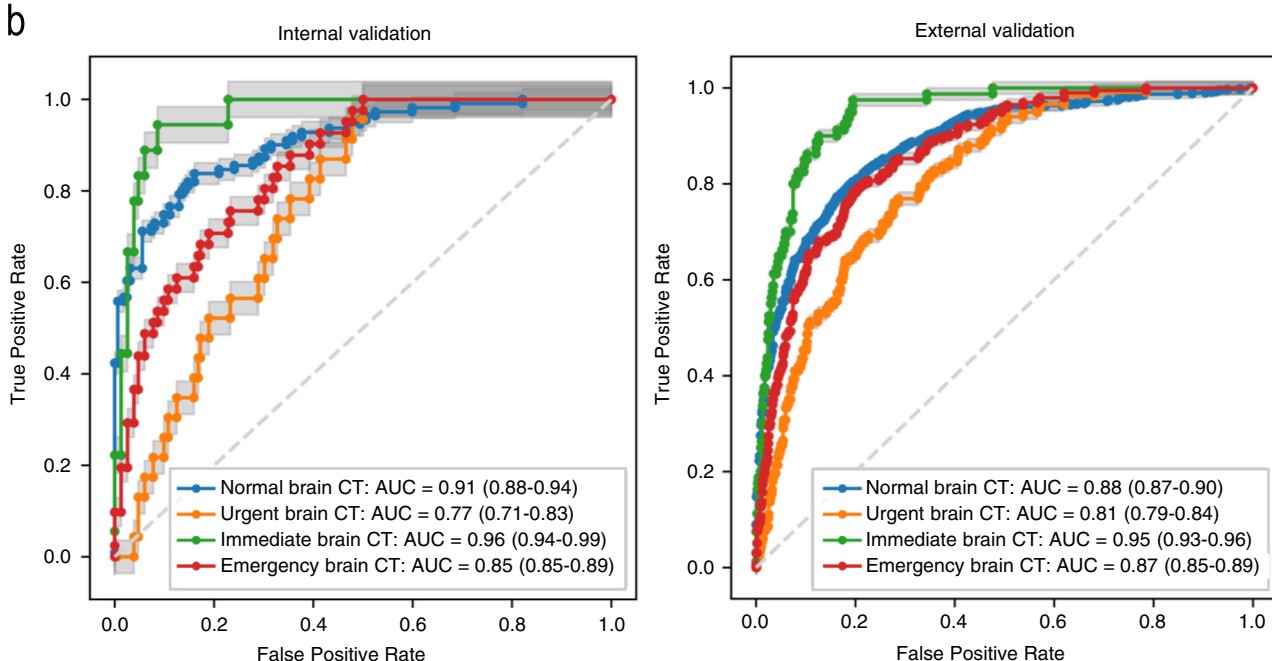

**Fig. 3 Detection performance of the ADA for brain CT triage. a** In both the internal and external validation tests, the anomaly scores differed significantly between non-emergency ($n = 232$ for internal validation; $n = 1598$ for external validation) and emergency cases ($n = 41$ for internal validation; $n = 197$ for external validation) (all $p < 0.001$). Box plots show the median (center line), first and third quartiles (box edges), and whiskers 1.5 times the IQR. Data points outside the whiskers are considered outliers. Two-sided $p$-value was calculated using independent $t$-tests. **b** ROC curve analysis for assessing the performance of the ADA according to different target groups in the internal and external validation tests. Date are presented as mean AUC values with 95% CI. Source data including exact $p$-values are provided in the Source Data file.

emergency brain CT scan. Unexpectedly, the increase in the WT and TAT in non-emergency cases was significantly smaller than the decrease in the WT and TAT in emergency cases. This finding is likely due to the small percentage of emergency cases and shorter RT following ADA implementation in non-emergency cases. Although the emergency cases led to a radiology workflow delay in the non-emergency cases, the faster RT in the relatively larger non-emergency cases seemed to offset these effects. Given our study design with a clinical simulation test, the shorter RT in the non-emergency cases may be due to the change in the radiologists' confidence or behavior for image interpretation in the normal brain CT scans predicted by ADA rather than due to recall bias or a learning effect. However, this issue needs further study.

The unresolved problem for anomaly detection models is the relatively high false-positive and false-negative rates. In the randomized controlled study conducted by Titano et al.[18], their supervised model for the triage of urgent brain CT scans could alert physicians in 50% of critical cases, with a 21% false-alarm rate. Our model had a high false-negative rate (22.3%) and false-positive rate (19.1%). In our clinical simulation test, the ADA implementation caused a significant delay in the median WT and TAT in the false negatives compared with the pre-ADA group. Therefore, the triage system with the anomaly detection model posed a risk of undermining the timely management of patients with critical CT findings. For false positives, a false alarm can reduce physicians' faith in a model and negatively affect emergency patients who need fast treatment. Although these problems

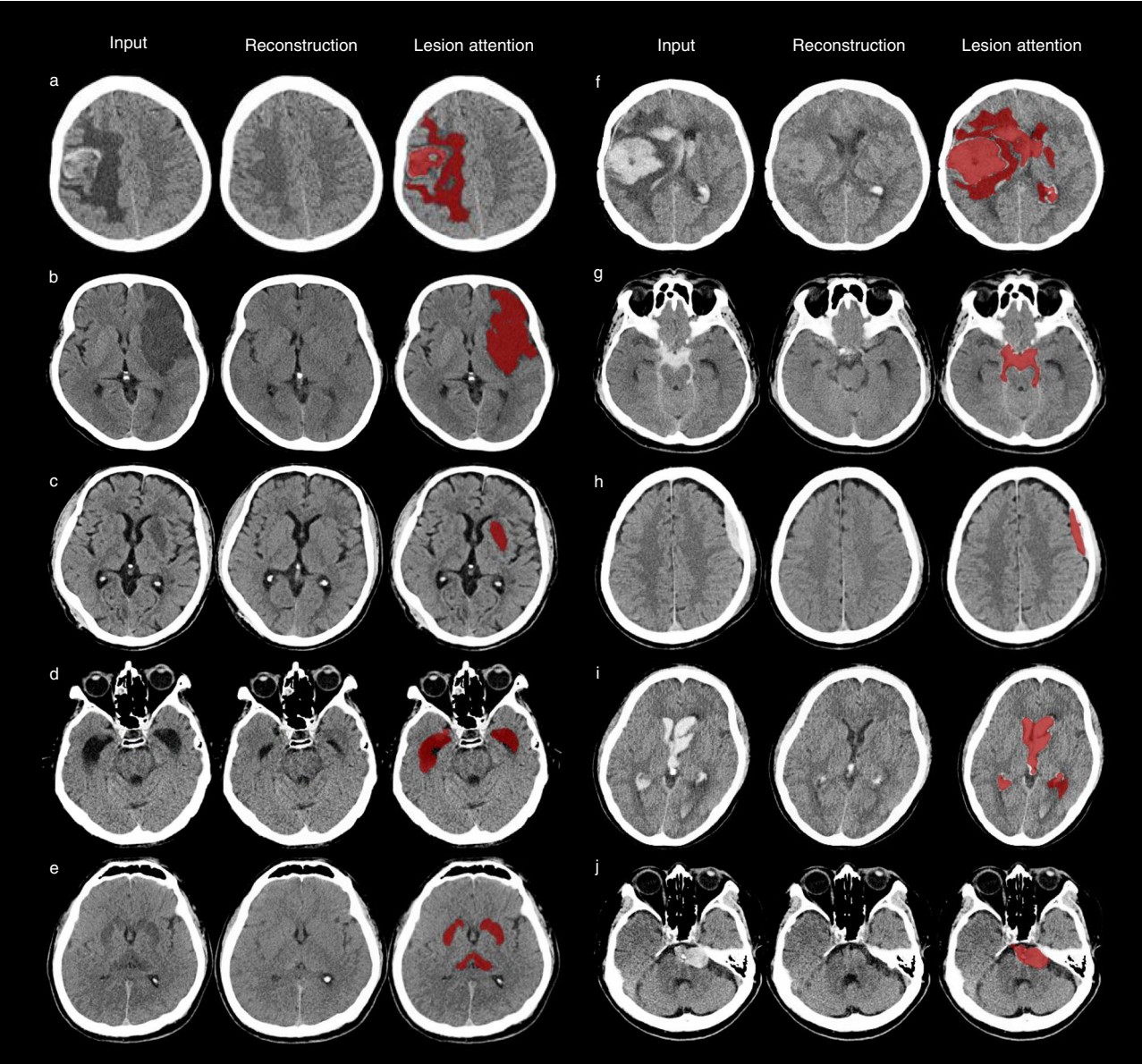

**Fig. 4 Localization of the predicted lesion on emergency brain CT images from patients with various diseases.** The columns, from the left to right, of each case represent input images, reconstructed images, and lesion attention. The attention maps localize anomalies related to secondary brain changes such as midline shift or perilesional edema as well as space-occupying brain lesions. **a** brain mass-like lesions, **b** acute territory infarction, **c** acute basal ganglionic infarction, **d** hydrocephalus, **e** hypoxic encephalopathy, **f** intracerebral hemorrhage (ICH), **g** subarachnoid hemorrhage (SAH), **h** subdural hemorrhage (SDH), **i** intraventricular hemorrhage (IVH), and **j** unruptured aneurysm.

could be solved using technical advances, this will be an ongoing issue unless the triage algorithm achieves perfect accuracy. Therefore, it is important that interpreting radiologists understand the optimization strategy and are prepared to deal with false positives or negatives.

This study has several limitations. First, our current system relies on a single brain CT scan and does not refer to prior imaging examinations or clinical information. This could result in mis-triage of some less urgent cases as high priority cases. For example, even if a previously diagnosed infarction has already been treated, it could be detected as an emergency case. Furthermore, anomaly cases of benign conditions (e.g., an arachnoid cyst or encephalomalacia with an old infarction) may also be incorrectly classified as emergency conditions. In addition, brain shrinkage is a normal part of the aging process but can indicate early-onset neurodegenerative diseases in younger patients.

Therefore, generating brain images that are the closest to normal without age information is challenging. Age information could be a prerequisite for correct classification in our anomaly detection model. These problems can be mitigated by training the model on benign conditions and incorporating meta-information regarding factors that affect clinical diagnosis. Third, we used clinical and radiological diagnoses as reference standards. However, many neurological ED cases (e.g., small traumatic intracranial hemorrhage, minor stroke, or transient ischemic attack) do not require surgical treatment or aggressive intervention because of their low risk of rapid exacerbation. Therefore, this may be an unavoidable limitation in an emergency screening cohort study. Nevertheless, further studies using the gold standard are warranted to determine the accurate performance of the model. Fourth, this study did not reflect the complexity of clinical practice. Multiple factors can influence the results of a clinical simulation test, including the

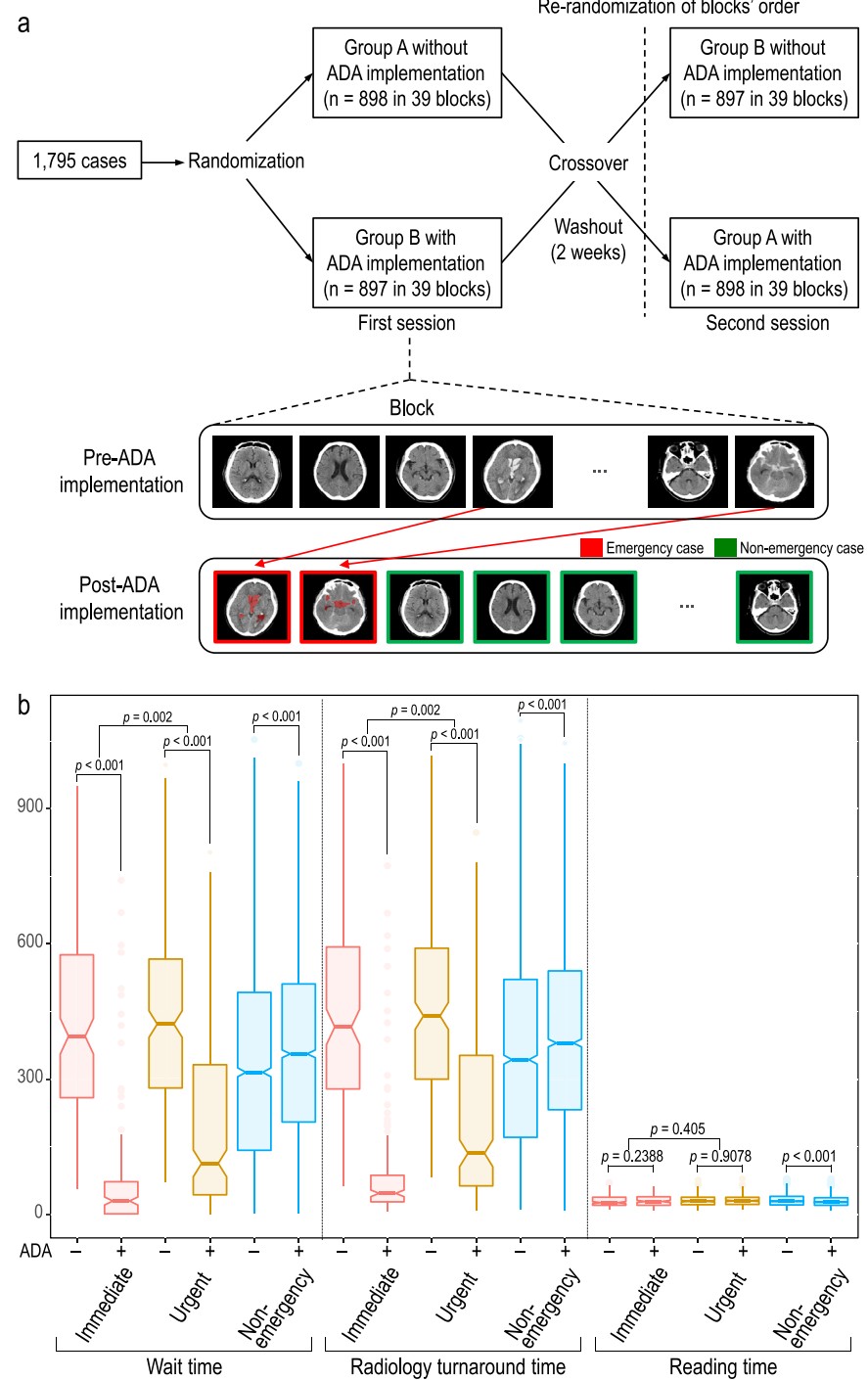

**Fig. 5 Clinical simulation test. a** Randomized crossover study design. **b** Comparison of outcomes in subgroups pre- and post-ADA triage (immediate [$n = 80$], urgent [$n = 117$], and non-emergency cases [$n = 1598$]). Data are reported as the median ± IQR. Box plots show the median (center line), first and third quartiles (box edges), and whiskers 1.5 times the IQR. Data points outside the whiskers are considered outliers. Two-sided $p$-values were calculated using the Wilcoxon signed-rank test for comparison between pre- and post-ADA triage, and the Wilcoxon rank-sum test was used for comparison between immediate and urgent cases. Source data including exact $p$-values are provided in the Source Data file. ADA anomaly detection algorithm, WT wait time, TAT radiology turnaround time, RT reading time.

case difficulty, queue size of the CT scan, readers' expertize level, image-processing time, patient acuity, and interruption by other examinations. Therefore, our results may vary with these factors. To address this issue, multicentered and prospective validation studies are warranted.

In conclusion, we developed an ADA with a deep generative network trained only on normal brain CT images from healthy individuals. Our model achieved moderate but consistent performance in detecting emergency brain CT scans using internal and external ED screening cohorts. In the clinical simulation test, our study also highlighted the feasibility of the ADA as a triage system to reprioritize radiology worklists and accelerate the diagnosis of various emergency conditions.

**Table 1 Comparison of outcomes pre-ADA and post-ADA triage.**

| | | | Pre-ADA | Post-ADA | Difference between pre- and post-ADA | p-value[a] | p-value[b] |
|---|---|---|---|---|---|---|---|
| Emergency (n = 197) | WT | Median (IQR) | 422.5 (299.0) | 70.5 (168.0) | −294.0 (352.0) | <0.001 | <0.001 |
| | | Mean (±SD) | 436.6 (±192.2) | 147.4 (±184.0) | | | |
| | | Min–Max | 1–997 | 1–803 | | | |
| | RT | Median (IQR) | 29.0 (12.5) | 30.0 (11.0) | 0.0 (13.0) | 0.38 | 0.006 |
| | | Mean (±SD) | 29.7 (±9.2) | 30.3 (±7.7) | | | |
| | | Min–Max | 9–76 | 7–79 | | | |
| | TAT | Median (IQR) | 445.0 (298.0) | 88.5 (179.0) | −297.5 (347.0) | <0.001 | <0.001 |
| | | Mean (±SD) | 457.9 (±195.4) | 168.7 (±183.2) | | | |
| | | Min–Max | 63–1017 | 6–847 | | | |
| Non-emergency (n = 1598) | WT | Median (IQR) | 327.0 (357.0) | 364.8 (307.4) | 79.3 (197.9) | <0.001 | |
| | | Mean (±SD) | 335.1 (±217.1) | 366.0 (±192.9) | | | |
| | | Min–Max | 1–1053 | 1–1000 | | | |
| | RT | Median (IQR) | 31.00 (11.5) | 28.00 (11.5) | −1.5 (14.0) | <0.001 | |
| | | Mean (±SD) | 31.2 (±8.9) | 29.7 (±9.2) | | | |
| | | Min–Max | 9–79 | 8–79 | | | |
| | TAT | Median (IQR) | 357.0 (352.0) | 393.0 (303.4) | 72.8 (202.3) | <0.001 | |
| | | Mean (SD) | 364.3 (218.0) | 393.2 (192.1) | | | |
| | | Min–Max | 12–1095 | 9–1045 | | | |

Data are expressed as the mean (SD, standard deviation) or median [interquartile range, IQR] (seconds). All statistical tests were two-sided, and statistical significance was set at p = 0.05.
[a]The Wilcoxon signed-rank test was used for comparison between pre- and post-ADA triage.
[b]The Wilcoxon rank-sum test was used for comparison between emergency and non-emergency cases.

**Table 2 Comparison of outcomes pre- and post-ADA triage among false negatives and false positives.**

| | | | Pre-ADA | Post-ADA | Difference between pre- and post-ADA | p-value[a] |
|---|---|---|---|---|---|---|
| False negatives (n = 44) | WT | Median (IQR) | 358.0 (291.5) | 449.8 (199.3) | 71.0 (145.0) | 0.009 |
| | | Mean (±SD) | 400.5 (±192.2) | 445.0 (±150.4) | | |
| | | Min–max | 72–922 | 146–803 | | |
| | RT | Median (IQR) | 28.8 (10.0) | 28.8 (9.4) | −0.3 (9.9) | 0.68 |
| | | Mean (±SD) | 29.4 (±8.6) | 29.6 (±7.8) | | |
| | | Min–max | 9–76 | 12–62 | | |
| | TAT | Median (IQR) | 384.3 (300.9) | 471.0 (205.0) | 70.3 (143.6) | 0.02 |
| | | Mean (±SD) | 421.7 (±196.2) | 464.5 (±150.9) | | |
| | | Min–max | 82–951 | 155–847 | | |
| False positives (n = 305) | WT | Median (IQR) | 357.0 (366.0) | 101.0 (104.0) | −220.5 (360.5) | <0.001 |
| | | Mean (±SD) | 342.9 (±220.4) | 111.1 (±76.8) | | |
| | | Min–max | 1–957 | 1–449 | | |
| | RT | Median (IQR) | 32.5 (11.5) | 35.5 (13.0) | 2.5 (14.5) | <0.001 |
| | | Mean (±SD) | 33.6 (±9.1) | 36.6 (±9.6) | | |
| | | Min–max | 9–79 | 9–79 | | |
| | TAT | Median (IQR) | 378.0 (357.5) | 134.50 (111.5) | −223.5 (361.5) | <0.001 |
| | | Mean (SD) | 374.0 (±220.8) | 143.7 (±78.1) | | |
| | | Min–max | 12–1007 | 9–508 | | |

Data are expressed as the mean (SD, standard deviation) or median [interquartile range, IQR] (seconds). All statistical tests were two-sided, and statistical significance was set at p = 0.05. [a]The Wilcoxon signed-rank test was used for comparison between pre- and post-ADA triage.

## Methods

**Ethics statement**. This retrospective study was conducted in accordance with the principles of the Declaration of Helsinki and current scientific guidelines. The Institutional Review Boards (IRBs) of Asan Medical Center (2019-0795) and Gangneung Asan Hospital (GNAH 2020-01-006) approved the study protocol. They waived the requirement for informed patient consent, given the minimal risk to subjects in the retrospective imaging study and the impracticality of obtaining informed consent from large numbers of patients retrospectively.

**Data collection, curation, and categorization**. For the development of CN-StyleGAN, a total of 197,038 non-contrast brain CT scans and paired radiology reports were retrospectively collected from patients who visited an urban, tertiary, academic hospital between January 1, 2000, and August 31, 2018. After iterations of the data curation process, the training dataset comprised 34,085 normal brain CT scans from healthy patients. In detail, the data curation process included three steps. First, we reviewed the radiology reports from 10,000 randomly sampled CT scans and selected keywords for anomalous CT findings such as positive pathological findings, benign lesions, and postoperative changes. Second, a natural language processing (NLP) algorithm (PyConTextNLP[20]) was used to exclude anomalous brain CT scans based on these keywords. Finally, two radiologists (GS Hong and B Jeong, with 14 years and four years of experience in reading brain CT images, respectively) randomly selected 1,000 CT scans and reviewed their radiology reports. If anomalous CT scans were found during this step, additional keywords were added. This data curation cycle was repeated five times to obtain completely normal CT scans. A total of 79,060 postoperative CT scans and 78,713 abnormal CT scans were excluded. Finally, the NLP-based data curation was assessed by manually reviewing the radiology reports of 1000 randomly selected cases. Of the 39,265 potentially eligible cases, CT scans from 5180 cases were not available for automatic downloading using the in-house system. Finally, the brain CT scans of 34,085 normal individuals were included in the training dataset.

Furthermore, the brain CT scans were collected independently and retrospectively from consecutive individuals who underwent emergency screening

for suspected neurological conditions in the EDs of an internal and an external institution. For the tuning and internal validation test, after six cases were excluded due to download errors, 544 non-contrast brain CT scans of ED patients were consecutively collected from Asan Medical Center in February 2019. The internal dataset was subsequently randomly divided into two parts: a tuning dataset and an internal validation dataset, and the ratio of each emergency severity group was preserved. For the external validation test, 1795 non-contrast brain CT scans from ED patients were consecutively collected from Gangneung Asan Hospital from January 1, 2019, to May 31, 2019. A board-certified emergency radiologist (GS Hong, with 14 years of experience reading brain CT images) reviewed all CT images in the internal and external validation datasets and classified the cases according to the category system for emergency severity[21–23]. This system categorized the cases into the following categories based on the urgency of treatment: normal, benign, indeterminate, urgent, and immediate. Subsequently, both urgent and immediate cases were defined as emergency cases. Cases of a critical, life-threatening condition that required immediate medical or surgical treatment were defined as immediate cases. Cases that were not life-threatening currently but required rapid treatment because they could deteriorate were defined as urgent cases. The disease entities in the emergency cases were categorized as brain mass-like lesions, acute infarctions, intracranial hemorrhages, hydrocephalus, and other diseases. A brain mass-like lesion was defined as a volumetric space-occupying lesion (e.g., brain tumor, brain abscess, tumefactive demyelinating disease, or encephalitis) distinct from the brain parenchyma with a normal appearance.

**Development of CN-StyleGAN**. We developed an architecture, termed CN-StyleGAN, that was closely modeled after StyleGAN2. CN-StyleGAN comprised three deep neural networks: a style-based generator (G), discriminator (D), and style-based encoder (E). We used the same architecture as StyleGAN2 for G and D; E followed the architecture of D, although the last fully connected layer was modified to output an 8192-dimensional latent code, $\mathbf{w} \in \mathbf{W}^+$, followed by a leaky ReLU of $\alpha = 0.2$[24]. Given a brain CT image as an input, E encodes the image into the closest normal-style latent code, and G generates the closest normal-style brain CT image from the latent code, trying to fool D by making the generated image indistinguishable from the true image. Then, D tries to discriminate the generated image from the true image.

**Training**. Supplementary Fig. 4 illustrates the training process of CN-StyleGAN. We trained CN-StyleGAN using normal brain CT images and several training processes for the model to encode the style of normal brain CT images. First, we trained G and D for 160,000 iterations following the original training process of StyleGAN2. Subsequently, we trained E and D but not G with loss functions including VGG16-based learned perceptual image patch similarity (LPIPS) loss[25,26], domain-guided loss[27], and adversarial loss functions from StyleGAN2. LPIPS loss measured the discrepancy between real images ($\mathbf{x}$) and reconstructed images (G(E($\mathbf{x}$))) in the feature space of VGG16. To improve the performance and increase the stability, we downsampled the images to a resolution of $256 \times 256$ pixels before computing the LPIPS distance. The domain-guided loss measured the L1 distance between E($\mathbf{x}$) and E(G(E($\mathbf{x}$))) for the in-domain property, regularizing the latent code to be inside the latent space of the normal brain CT data distribution. For adversarial loss, non-saturating loss[28] was used with R1-regularization[29] at every 16th step to stabilize the training of D. After adversarial training, the reconstructed images were indistinguishable from the normal brain CT images. Furthermore, random erasing of brain CT images[30] was used so that E could learn the semantics of normal brain CT images by filling in the missing region. We trained the model in PyTorch[31] with the Adam optimizer[32] for 200,000 iterations with hyper-parameters ($\beta_1 = 0$, $\beta_2 = 0.99$, $\varepsilon = 10^{-8}$, and minibatch = 32). The learning rate was $10^{-5}$ for the E and $10^{-6}$ for the D.

**Gaussianized latent space**. Previous studies on StyleGAN have indicated that data distribution can be explicitly modeled as a normal distribution in the intermediate latent space of StyleGAN[33,34]. Similarly, we explicitly modeled the data distribution of normal brain CT images in the intermediate latent space. We used E to map each normal brain CT image, slice-by-slice, from the training data to the latent space and used the latent codes to estimate the sample statistics for each slice order. Thus, the empirical covariance matrices, $\boldsymbol{\Sigma}$, and means, $\boldsymbol{\mu}$, were accumulated for each layer of the intermediate latent space.

**Inference**. Supplementary Fig. 5 illustrates the inference method and anomaly scoring system of CN-StyleGAN. A CT scan included up to 32 axial slices from the bottom to the top. We initialized the latent code, $\mathbf{w_{init}}$, for each axial slice ($\mathbf{x}$) of the scan as E($\mathbf{x}$) and the noise maps ($\mathbf{n}$) from a normal distribution. We Gaussianized and optimized the latent code ($\mathbf{w}$) with L1, LPIPS, and the in-domain loss functions using the Adam optimizer for 100 epochs. Furthermore, the in-domain loss was modified to regularize the latent vector in the Gaussianized latent space only when the latent code deviated from the mean of the data distribution of normal brain CT images in the latent space compared with the in-domain latent code, E(G($\mathbf{x}$)). After the latent code was optimized as $\mathbf{w}^*$, we optimized the noise maps with the L1 loss function for 100 iterations. Noise maps can be optimized to generate out-of-

domain images[35]; therefore, we proposed a masked noise optimization that forced the model to reconstruct the normal region alone. At each optimization step, a binary mask, $\mathbf{M}$, was defined to predict the lesion area in the scan. To calculate $\mathbf{M}$, the residual difference between an image ($\mathbf{x}$) and the reconstructed image (G($\mathbf{w}^*$, $\mathbf{n}$)) was brain-extracted[36], median-filtered with a window size of 17, and thresholded by 5 Hounsfield units. Moreover, the number of false positives in $\mathbf{M}$ decreased because of the intersections of binary masks at the previous optimization steps. Consequently, $\mathbf{M}$ was used to set a target image for optimization:

$$\mathbf{x}_{target} = \mathbf{M} \odot G(\mathbf{w}^*, \mathbf{n}_{init}) + (1 - \mathbf{M}) \odot \mathbf{x} \quad (1)$$

where $\odot$ denotes a pointwise multiplication. At the last optimization step, the binary mask was used as the lesion attention map for prediction.

**Anomaly score**. The anomaly score was calculated as follows: first, reconstruction error for a slice $\mathbf{x}_i$ of a scan was defined as:

$$R(\mathbf{x_i}) = \| \mathbf{M} \odot (\mathbf{x_i} - G(\mathbf{w}, \mathbf{n})) \| \quad (2)$$

which is the binary masked density error between the slice, $\mathbf{x}_i$, and the reconstructed slice, G($\mathbf{w}$,$\mathbf{n}$). Second, this reconstruction error was normalized, slice-by-slice, based on the slice order, using the reconstruction error statistics of the mean, $\mathbf{R}_\mu$, and SD, $\mathbf{R}_\sigma$, of the normal brain CT images. A total of 1000 scans were randomly selected from the training dataset for the normal brain reconstruction error statistics. Finally, this normalized per-slice reconstruction error of 32 slices for the scan was summed to obtain the anomaly score:

$$\text{Anomaly score} = \sum_{i=1}^{32} \frac{R(\mathbf{x_i}) - \mathbf{R}_{\mu_i}}{\mathbf{R}_{\sigma_i}} \quad (3)$$

**Clinical simulation test**. To investigate the effect of CN-StyleGAN-aided radiology workflow reprioritization, we retrospectively performed a clinical simulation test using the external validation dataset of 1795 brain CT scans. Two radiologists (WJ Jung and JH Lee, each with ≥14 years of experience in reading brain CT images) independently and retrospectively performed a clinical simulation test using a washout period and varying reading orders in a crossover design to assess brain CT scans with and without the help of the triage system. Specifically, a total of 1795 brain CT scans from the external validation dataset were randomized to two groups (group A [898 brain CT scans] and group B [897 radiographs]). Each block enrolled 23 brain CT scans, except for one block in group A that enrolled 24 brain CT scans, as the number of imaging studies ($n = 878$) in group A could not be divided evenly by 23. In the first session, each reader assessed the brain CT scans in group A without the help of the triage system and those in group B with the help of the triage system. In the second session, each reader assessed the brain CT scans in group A with the help of the triage system and those in group B without the help of the triage system. The first and second sessions were separated by at least two weeks, and the reading order of the blocks was randomized and different for each reading session. Our triage system reprioritized emergency cases based on their anomaly scores and labeled them in red in the worklist to attract the readers' attention. The readers were able to overlay the segmentation mask (lesion attention) predicted by CN-StyleGAN on the brain CT image. The readers interpreted the brain CT images and determined the presence of critical findings in the CT scans using an in-house user interface that provided the radiology worklists of the brain CT scans and their images (Supplementary Fig. 6). The readers were blinded to the clinical information, imaging reports, and number of emergency cases included in the study.

Three radiological time metrics, including WT, TAT, and RT, were selected based on previous studies[6,15,16]. These time metrics were calculated based on the timepoints in the CT interpretation process, which were automatically recorded by the software. The metrics were calculated for each case in each block and were defined as follows:

$$\text{WT}(i) = (\text{Timestamp of opening CT}_i) - (\text{Timestamp of opening a block}) \quad (4)$$

$$\text{TAT}(i) = (\text{Timestamp of reporting image findings in CT}_i) - (\text{Timestamp of opening a block}) \quad (5)$$

$$\text{RT}(i) = (\text{Timestamp of closing CT}_i) - (\text{Timestamp of opening CT}_i), \quad (6)$$

where $\text{CT}_i$ is the $i$ th CT in a block.

**Statistical analyses**. The mean values of the anomaly scores between emergency and non-emergency cases were compared using independent $t$-tests. The emergency case detection performance of CN-StyleGAN was analyzed by calculating the AUC, sensitivity, specificity, and accuracy in the internal and external validation datasets. The optimal anomaly score cutoff value was determined from the maximum value of Youden's index for the ROC curve analysis using the tuning dataset. The bootstrap method (10,000 iterations) was used to calculate 95% CIs. The median values of the time metrics in the clinical simulation test were compared using the Wilcoxon signed-rank test and Wilcoxon rank-sum test. Analyses were performed using Python version 3.8.5 (sklearn 0.23.2; Python Software

Foundation), R version 4.1.0 (R Foundation for Statistical Computing), and ggplot2 version 3.6.3. All statistical tests were two-sided, and the statistical significance was set at $p = 0.05$.

**Reporting summary**. Further information on research design is available in the Nature Research Reporting Summary linked to this article.

## Data availability

The raw experimental and clinical data are provided as Source Data, including the diagnoses (ground truths), the model-prediction anomaly scores and detection performance (Fig. 3), and the clinical simulation results for wait time, radiology turnaround time, and reading time (Fig. 5, Tables 1 and 2). The brain CT images for the development and validation of the model are not publicly available because they contain private patient health information. For reasonable purposes including reproducing results in this study, researchers can request the corresponding authors, G.-S.H. and N.K., with approval of the Institutional Ethics Committee of Asan Medical Center. The requests will be processed in 60 business days. Source data are provided with this paper.

## Code availability

All code related to this project was written in Python. Custom code for image extraction, the pre-processing pipeline, the deep learning model builder, the data provider, and the experimenter driver is available at https://github.com/seungjunlee96/emergency-triage-of-brain-computed-tomography-via-anomaly-detection-with-a-deep-generative-model.

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

## Acknowledgements

This research was supported by a grant from the Korea Health Technology R&D Project through the Korea Health Industry Development Institute (KHIDI), funded by the Ministry of Health & Welfare, Republic of Korea (HI21C1148 to G.S.H.). The authors are grateful to Hyun-Jin Bae, Ph.D of Promedius Inc. for the technical support. The authors are grateful to Ju Hee Lee, MD, Ph.D for participating as a reader and Sehee Kim in Department of Clinical Epidemiology and Biostatistics, Asan Medical Center, for performing statistical analyses.

## Author contributions

G.S.H. and N.K. designed the study; J.K., W.P., and G.S.H. collected data; B.J., G.S.H., and M.K. performed data curation; S.L., M.K., R.J., and N.K. developed the deep learning algorithms; G.S.H. designed and supervised the clinical simulation test; W.J.C. and B.J. performed the clinical simulation test; S.L. and G.S.H. performed the statistical analyses; all authors contributed to the interpretation of data and writing and editing the manuscript.

## Competing interests

The authors declare no competing interests.
