## [Peer Review File · Nature Communications]

Reviewers' Comments:

Reviewer #1:

Remarks to the Author:

Authors present an anomaly detection method based on neural networks focusing on a very interesting application. They develop and present an explainable anomaly detection method for brain CT. The main application is triage in an emergency department (ED). The ultimate goal is to reduce the diagnosis and reporting time of cases with critical findings in their brain CT. They report that the method can reduce the median diagnosis start time from 4 minutes to close to 40 seconds, which is impressive. The clinical value of this reduction is not explicitly supported but it is assumed.

Technology-wise the proposed method is based on existing technology. Therefore, the novelty is limited.

Application-wise the article focuses on an application similar to what has been done in [16]. Despite some notable differences, i.e., external validation, there is a large overlap and thus the novelty in application is also not very high.

The single biggest novelty of the presented article is the validation. Different than other similar works in the literature, authors here used two different validation datasets and used - what appears to be - clinically relevant end points, time to diagnose and report all critical cases in a block of 20 CT scans. Furthermore, they also used datasets comprised of images acquired with scanners from different manufacturers.

That said, both the method and experiments raises some concerns. I am listing the major ones below.

Methods:

The explainable component of the model is over emphasized and not demonstrated. First of all, a very large majority of anomaly detection methods in the literature not only detect abnormal scans but also locate the anomaly in the images. For instance the article cited as [11] is an example to this end. The method proposed here seems to be the same thing. Thus, I am not clear about the novelty in this respect. Secondly, the value of the "explainable" part of the method is not demonstrated in the experiments. If anything, it looks like RT (reading time) does not change when the method is used. Authors mention in the discussion that "... it is notable that the model's prediction on the severity is easily understandable as it is based on the density error of the detected lesion." This is a very vague statement without any support in the article. If the "explainable" part of the model is crucial and I am missing something then I encourage authors to demonstrate it. If not, then I suggest not emphasizing that part.

Experiments:

During the clinical simulations, it is unclear whether random and reprioritized experiments were done on the same set of samples or not. Did the experts see the same samples twice?

Results and discussions:

Sensitivity of the method is around 0.85 and 0.80 for the internal and external validation sets, respectively. For this application, I believe this is a critical number. More specifically, given that patients in the emergency

group require immediate attention, I believe characterization of the model at 100% or close to 100% sensitivity is important. Accordingly, an important metric to additionally report is the False Positive Rate at 100% sensitivity level.

Explanations of the results can improve. Meanings of DST, RRT and RT are not perfectly clear for this reviewer. As these metrics are crucial for this article, their explanation should be very clear. Assuming DST and RRT refers to starting diagnosis and reporting critical findings of all CT in a group in a block, respectively, then

- I cannot understand how the proposed method reduces DST and RRT for both the emergency and non-emergency groups. This sounds like a zero-sum game. If DST and RRT decreases for one of the groups, it should increase for the other one.

- While reporting median times is reasonable, I believe it is important to also report worst case performance of the method for this application. More specifically, the range [min,max] of DST and RRT values across the blocks should be reported. My reasoning is as follows: if the method's worst case performance is worse than the worst case performance of the random triage, that means at least one emergency patient would receive attention later than they would in random assignment. This should then be taken into account for any clinical translation of the proposed method.

Reviewer #2:

Remarks to the Author:

The authors present a very interesting concept of leveraging generative adversarial networks to generate normal-appearing synthetic images based on real non-contrast head CT images from patients with a variety of diseases. By comparing the differences between real and synthetic normal-appearing images, the pipeline generates an "attention map" that highlights the abnormality and computes an anomaly score which is used to drive worklist prioritization. A few reports have described similar approaches, including a paper I recommend as a reference – "Modeling Healthy Anatomy with Artificial Intelligence for Unsupervised Anomaly Detection in Brain MRI" by Baur et al. However, the authors go beyond and test the tool in a simulated clinical testing to assess the benefit of such tool in the identification and reporting of urgent/emergent diseases. I think this paper is overall an important contribution to the field. However, there are several issues that require attention:

There are several misspellings throughout the paper (e.g. "Patent-level performance of CN-StyleGAN", "showed expert-level detection performance but were to focus on detecting a single or a few specific diseases"). Also, several phrases are difficult to understand and the manuscript would benefit from a grammatical review. Additionally, the authors should refrain from praising their own work and let the reader be the judge of the quality of the paper (e.g. "Our rigorous study design – including high-quality internal and external validation datasets that were...").

When comparing the study with research from other groups, the authors should be more objective with their comments. Comments such as "Lack of accurate measurement and careful validation of data result in failure of the reliability of a study." seem very subjective when no specific reasoning or supporting arguments are given. Also, a direct performance comparison with other publications ("However, it had an inferior performance as compared with our model") is difficult, unless the datasets were constructed in a reasonably similar fashion. This brings up an important point: Why were the artifacts (9 from the internal validation and 35 from the external validation sets removed? This inappropriately inflates performance and is my main criticism of this paper. In my view, without these scans, the performance reported cannot be considered an estimate of real-world performance. Unless the algorithm has mechanisms to detect these problematic scans and report them separately. This was not discussed in the paper.

Another criticism is that the algorithm is more of an anomaly detection than a critical finding identification system. This has to be stated as a limitation of the algorithm and not an issue of the validation set as stated in the manuscript "While our model did not achieve extremely accurate

performance, this may be due to our validation dataset that included many anomalous cases but not requiring emergency intervention (for example, encephalomalacia, severe leukoaraiosis, or arachnoid cysts in the false positives).” I also disagree with the statement that false positives are not a problem for triage systems. They increase noise, decrease reliability of the system, and may impact trust. Also, regarding the following statement: “Moreover, the strength of our method, a triage tool that is explainable and generally applicable for various diseases, alleviates the need for achieving extremely accurate performance.” I agree that explainability is an extremely helpful feature and may expedite review, but I am not sure that the strength of the method (which again should be judged by the reader) should alleviate issues with performance.

Another statement that seems inappropriate is the following: “However, in diagnosing most cases with emergency neurologic lesions, a typical image finding is pathognomonic by itself.” I do not think this is true. Imaging findings are rarely pathognomonic, they are typically associated with differential diagnosis, and not having prior imaging to compare makes this even more complicated. I also think that not considering relevant prior imaging studies as part of the decision making to determine whether a study is emergent or not is a limitation that needs to be clearly stated.

And finally, the authors should expand on the anomaly score system and provide examples. A detailed explanation of how the cutoff of the anomaly score was derived should be presented. Was this performed during tuning or validation?

Point-to-point responses to the reviewer's comments

Manuscript ID: NCOMMS-21-32482A

Title: “Emergency triage of brain computed tomography via anomaly detection based on a deep generative model”

We thank the reviewers for their suggestions and insightful comments. Regarding the concerns raised by the reviewers, we have carefully addressed the concerns and point-by-point responses to all comments are provided below. The title and abstract have been revised to meet the word limits. In addition, the manuscript has been edited for language, grammar, and clarity. Kindly note that the significantly revised sections in the revised manuscript and supplementary material are highlighted in yellow for the reviewers' convenience.

Reviewer #1

Authors present an anomaly detection method based on neural networks focusing on a very interesting application. They develop and present an explainable anomaly detection method for brain CT. The main application is triage in an emergency department (ED). The ultimate goal is to reduce the diagnosis and reporting time of cases with critical findings in their brain CT. They report that the method can reduce the median diagnosis start time from 4 minutes to close to 40 seconds, which is impressive. The clinical value of this reduction is not explicitly supported but it is assumed.

Technology-wise the proposed method is based on existing technology. Therefore, the novelty is limited. Application-wise the article focuses on an application similar to what has been done in [16]. Despite some notable differences, i.e., external validation, there is a large overlap and thus the novelty in application is also not very high. The single biggest novelty of the presented article is the validation. Different than other similar works in the literature, authors here used two different validation datasets and used - what appears to be - clinically relevant end points, time to diagnose and report all critical cases in a block of 20 CT scans. Furthermore, they also used datasets comprised of images acquired with scanners from different manufacturers.

That said, both the method and experiments raises some concerns. I am listing the major ones below.

1. **Methods:** The explainable component of the model is over emphasized and not demonstrated. First of all, a very large majority of anomaly detection methods in the literature not only detect abnormal scans but also locate the anomaly in the images. For instance the article cited as [11] is an example to this end. The method proposed here seems to be the same thing. Thus, I am not clear about the novelty in this respect. Secondly, the value of the "explainable" part of the method is not demonstrated in the experiments. If anything, it looks like RT (reading time) does not change when the method is used. Authors mention in the discussion that "... it is notable that the model's prediction on the severity is easily understandable as it is based on the density error of the detected lesion." This is a very vague statement without any support in the article. If the "explainable" part of the model is crucial and I am missing something then I encourage authors to demonstrate it. If not, then I suggest not emphasizing that part.

Author's response: We agree that the explainable component of the model was over emphasized and therefore, according to your suggestion, we have edited the title, abstract, and main text (*introduction* and *discussion*). The title has been changed from "An explainable triage on brain

computed tomography scans via anomaly detection using a deep generative model trained with normal scans” to “Emergency triage of brain computed tomography via anomaly detection based on a deep generative model”. In addition, we have deleted the word “explainable” from most instances in the main article. Moreover, we agree that the majority of anomaly detection methods in the literature not only detect abnormal scans but also locate the anomaly in the images, and therefore, we have added this comment.

Examples)

- “The model provided not only lesion localization but also the severity score of query scans in an explainable manner” (deleted)
- “an explainable triage system” has been revised to “a triage system”
- “an explainable anomaly detection algorithm” has been revised to “an anomaly detection algorithm”
- “Moreover, the anomaly detection framework based on deep generative models can visually highlight the model’s prediction.”

(Page 3, Line 36 – 37 in the Introduction)

2. Experiments: During the clinical simulations, it is unclear whether random and reprioritized experiments were done on the same set of samples or not. Did the experts see the same samples twice?

Author’s response: We apologize for the lack of clarity in the text. Two radiologists independently interpreted the same test set twice in two separate reading sessions in a blinded manner. There was an interval of 2 weeks between two reading sessions to avoid recall bias. In the revised manuscript, we have clarified this issue as follows:

In the revised manuscript:

- “Subsequently, two radiologists independently interpreted the same test set twice in two separate reading sessions in a blinded manner. There was an interval of 2 weeks between two reading sessions to avoid recall bias.”

(Page 7, Line 120 – 122 in the Result)

3. Results and discussions: Sensitivity of the method is around 0.85 and 0.80 for the internal and external validation sets, respectively. For this application, I believe this is a critical number. More specifically, given that patients in the emergency group require immediate attention, I believe characterization of the model at 100% or close to 100% sensitivity is important. Accordingly, an important metric to additionally report is the False Positive Rate at 100% sensitivity level.

Author's response: Thank you for the suggestion. Accordingly, we have mentioned the specificity values (1 – false positive rate) at 100% and 95% sensitivity levels in the main text.

In the revised manuscript:

- “We calculated the specificity (95% CI) at 0.95 and 1.00 sensitivity levels in the internal and external validation tests using the thresholds derived from the internal validation. At a sensitivity level of 0.95 in the internal validation test, the sensitivity and specificity of the internal validation test were 0.95 (0.90–0.99) and 0.55 (0.50–0.59), respectively, and those of the external validation test were 0.88 (0.83–0.92) and 0.63 (0.60–0.65), respectively. At a sensitivity level of 1.00 in the internal validation, the sensitivity and specificity in the internal validation test were 1.00 (1.00–1.00) and 0.26 (0.22–0.30), respectively, and those in the external validation test were 0.98 (0.96–1.00) and 0.35 (0.32–0.37), respectively.”

(Page 6, Line 107 – 114 in the Result)

4. Explanations of the results can improve. Meanings of DST, RRT and RT are not perfectly clear for this reviewer. As these metrics are crucial for this article, their explanation should be very clear.

Author's response: Thanks for your valuable comment. To clarify the meaning of the outcomes (time metrics), we have revised the terms of time metrics to those widely accepted in the radiology field. We have changed “diagnosis start time (DST)” to “wait time (WT)” and “radiology report time (RRT)” to “radiology report turnaround time (TAT).” Moreover, we have added the clinical meaning and calculation formulas of the metrics.

In the revised manuscript:

- “The clinical efficacy of the model was analyzed using three radiological time metrics based on previous studies¹²⁻¹⁴: WT, radiology report turnaround time (TAT), and reading time (RT) for each case in each block. A reduction in WT meant that radiologists had earlier access to brain CT scans for reading. A decrease in the TAT meant that the image findings were reported faster to the referring clinicians by radiologists. RT referred to the turnaround time from the start to the end of reading a CT scan. The WT and TAT should be as short as possible, especially for emergency cases.”

(Page 7, Line 126 – 132 in the Results)

- “Three radiological time metrics, namely WT, TAT, and RT, were selected based on previous studies¹²⁻¹⁴. These time metrics were calculated based on the time points in the CT interpretation process, which were automatically recorded in the software. The metrics were calculated for each case in each block and defined as follows:

$$\mathbf{WT}(i) = \mathbf{Timestamp\ of\ opening\ } CT_i \mathbf{ - Timestamp\ of\ opening\ a\ Block}$$

$$\mathbf{TAT}(i)$$

$$\mathbf{= Timestamp\ of\ reporting\ image\ findings\ in\ } CT_i \mathbf{ - Timestamp\ of\ opening\ a\ Block}$$

$$\mathbf{RT}(i) = \mathbf{Timestamp\ of\ closing\ } CT_i \mathbf{ - Timestamp\ of\ opening\ } CT_i$$

where CT_i is the CT image of i th case in the worklist of a block.”

(Page 16 – 17, Line 362 – 366 in the Methods)

5. Assuming DST and RRT refers to starting diagnosis and reporting critical findings of all CT in a group in a block, respectively, then - I cannot understand how the proposed method reduces DST and RRT for both the emergency and non-emergency groups. This sounds like a zero-sum game. If DST and RRT decreases for one of the groups, it should increase for the other one.

Author’s response: Thank you for the valuable comment. We partially agree that if WT and TAT decrease for one of the groups, it should increase for the other one. Ironically, these time metrics decreased in non-emergency cases, although they were read later. Given the small reduction of RT in both the non-emergency and emergency groups, it is reasonable to presume that the triage system’s reprioritization of the worklist and visualization of lesion attention cause behavioral

changes in radiologists, which was also observed in another study on a triage system. Nonetheless, the small decrease in WT and TAT of non-emergency cases may be clinically insignificant.

- “It is noteworthy that the proposed triage system can reprioritize emergency cases, leading to the reduction of WT and TAT. This result means that this system could be one of the strategies to improve workflow efficiency in neurological emergencies by reducing wait times for radiology reporting and acute care. However, there are a number of issues that need to be resolved before actual clinical application. Ironically, these time metrics decreased in non-emergency cases, although they were read later. Given the small reduction of RT in both the non-emergency and emergency groups, it is reasonable to presume that prioritization causes behavioral changes in radiologists, which was also observed in another study on a triage system¹⁶. It appears to be some type of automation bias^{19,20}, wherein readers over-rely on the software recommendation, which may then decrease the RT and subsequently affect WT and TAT. Nevertheless, compared with that in the emergencies, the degree of these reductions in WT and TAT was relatively more minor and thus may be clinically insignificant. Thus, we think this automation bias has minimal influence on the triage algorithm-driven positive effect in the present study.”

(Page 9, Line 178 – 190 in the Discussion)

6. While reporting median times is reasonable, I believe it is important to also report worst case performance of the method for this application. More specifically, the range [min,max] of DST and RRT values across the blocks should be reported. My reasoning is as follows: if the method's worst case performance is worse than the worst case performance of the random triage, that means at least one emergency patient would receive attention later than they would in random assignment. This should then be taken into account for any clinical translation of the proposed method.

Author's response: We agree with the reviewer's concern that the proposed method has potential risks of delayed reporting of false negatives. Given the limited resources in the emergency setting, it could be a zero-sum game because the gain of some patients is offset by the loss of others. Unless artificial intelligence-based triage has perfect accuracy, the problems will not be solved. This concern has been advanced by the previous works on handling the triage system by artificial intelligence. In addition, the automation bias might cause the radiologists to over-rely on the software recommendation and overlook the false-negative cases. Although the delay in WT and TAT of the false-negative cases was not statistically significant in our clinical simulation test, the clinical relevance of our triage system for the outcomes should be carefully

interpreted by measuring its net value. However, the retrospective study design has limitations in reflecting clinical practice complexity. Therefore, multicentered and prospective validation studies are warranted. We have added this problem in the limitation of the revised manuscript.

Hence, the distribution of data is important to interpret our results. According to the suggestion of the reviewer, all details (including min and max values) of our results have been summarized in Supplementary Table 4. In the case of outliers in the sample, the median and interquartile range are used to summarize a typical value and variability. According to the comment of a statistician, the results in the main text are presented as median and interquartile ranges. Instead, in the case of false negatives, the detailed data for delayed wait time (WT) and radiology report turnaround time (TAT) have been added. Please note that DST and RRT have been changed to WT and TAT, respectively, in the revised manuscript.

In the revised manuscript:

- “The outcomes of the clinical simulation test are presented as median values in seconds (interquartile range [IQR]). In the emergency group, the triage significantly decreased WT from 243 s (IQR: 112–366 s) to 42 s (IQR: 3–125 s) ($P < .001$) and TAT from 262 s (IQR: 134–392 s) to 63 s (IQR: 23–145 s) ($P < .001$). In the non-emergency group, a significant difference was found in WT (220 s [IQR: 103–352 s] vs. 209 s [IQR: 115–309 s], $P = .002$) and TAT (244 s [IQR: 124–380 s] vs 228 s [IQR: 136–328 s], $P < .001$) between two sessions (Figure 5b). Unfortunately, in false-negative cases, a lag in WT (from 235.5 s [IQR: 116.5–369.5 s] to 254.0 s [IQR: 185.0–353.0 s], $P = .22$) and TAT (from 258.0 s [IQR: 140.5–386.5 s] to 272.5 s [IQR: 205.0–379.0 s], $P = .30$) was noted, but without statistical significance. In the non-emergency and emergency groups, a significant difference was noted in the RTs between two reading sessions (22 s [IQR: 17–30 s] vs 17 s [IQR: 14–23 s], $P < .001$; 18.0 s [14–27 s] vs. 18.0 s [14–27 s], $P < .001$). In the case of false negatives, no significant difference was noted in RT (20.5 s [IQR: 15.0–18.0 s] vs. 19.0 s [IQR: 13.0–25.5 s], $P = .07$) (Supplementary Table 4).”

(Page 7, Line 133 – 145 in Results)

- **Supplementary Table 4.** Comparison of time metrics between groups without and with triage using CN-StyleGAN

		Non-emergency CT group			Emergency CT group		
		Control	Reprioritized	P	Control	Reprioritized	P
WT	Median	220	209	.002	243	42	< .001

	(IQR)	(103–352)	(115–309)		(112–366)	(3–125)	
	min–max	1–1360	1–1176		1–1313	1–1114	
TAT	Median	244	228	< .001	262	63	< .001
	(IQR)	(124–380)	(136–328)		(134–192)	(23–145)	
	min–max	10–1398	9–1230		20–1320	3–1131	
RT	Median	22	17	<.001	18	18	< .001
	(IQR)	(17–30)	(14–23)		(14–27)	(10–27)	
	min–max	8–503	4–231		5–178	3–495	

Data are presented as the median value (seconds) with interquartile range (IQR) or minimum and maximum values in parentheses. Abbreviations: CT, computed tomography; WT, wait time; RT, reading time; TAT, radiology report turnaround time.

- “However, our results showed that this triage system has a risk of delayed reporting of false negatives, although this result was not statistically significant and the change in time metrics was relatively smaller in false-negative cases than in emergency cases. Therefore, given the limited resources in the emergency setting, it could be a zero-sum game because the gain in some patients is offset by the loss in other patients.”

(Page 9, Line 190 – 194 in the Discussion)

- “First, our triage system has potential risks of delayed reporting of false negatives as well as detecting false positives. Further work is required to reduce these risks.”

(Page 10, Line 207 – 208 in the Discussion)

Reviewer #2

The authors present a very interesting concept of leveraging generative adversarial networks to generate normal-appearing synthetic images based on real non-contrast head CT images from patients with a variety of diseases. By comparing the differences between real and synthetic normal-appearing images, the pipeline generates an “attention map” that highlights the abnormality and computes an anomaly score which is used to drive worklist prioritization.

A few reports have described similar approaches, including a paper I recommend as a reference – “Modeling Healthy Anatomy with Artificial Intelligence for Unsupervised Anomaly Detection in Brain MRI” by Baur et al. However, the authors go beyond and test the tool in a simulated clinical testing to assess the benefit of such tool in the identification and reporting of urgent/emergent diseases. I think this paper is overall an important contribution to the field.

Author’s response: Thank you for your positive comment. According to your suggestion, we have added the recommended paper in our reference list.

1. However, there are several issues that require attention: There are several misspellings throughout the paper (e.g. “Patent-level performance of CN-StyleGAN”, “showed expert-level detection performance but were to focus on detecting a single or a few specific diseases”. Also, several phrases are difficult to understand and the manuscript would benefit from a grammatical review.

Author’s response: We thank the reviewer for pointing out grammatical issues. Accordingly, we have carefully reviewed the entire paper and all spelling and grammatical errors indicated by reviewers #1 and #2 have been corrected. In addition, the manuscript has been updated by additional professional English editing service.

2. Additionally, the authors should refrain from praising their own work and let the reader be the judge of the quality of the paper (e.g. “Our rigorous study design – including high-quality internal and external validation datasets that were...”). When comparing the study with research from other groups, the authors should be more objective with their comments. Comments such as “Lack of accurate measurement and careful validation of data result in failure of the reliability of a study.” seem very subjective when no specific reasoning or supporting arguments are given.

Also, a direct performance comparison with other publications (“However, it had an inferior performance as compared with our model”) is difficult, unless the datasets were constructed in a reasonably similar fashion.

Author’s response: We appreciate the reviewer’s insightful suggestion. We have edited the entire manuscript according to your suggestion.

Examples)

- “Lack of accurate measurement and careful validation of data result in failure of the reliability of a study.” and “However, it had an inferior performance as compared with our model.” (deleted).
 - “e.g. robust triage system” (deleted)
 - The original comment “Our rigorous study design – including high-quality internal and external validation datasets that were...” has been revised as follows: “We developed an anomaly detection algorithm based on a deep generative model trained with brain CT images of healthy individuals as a triage system. The algorithm was validated using the data of the internal and external validation datasets, which included consecutive emergency patients who underwent screening in the ED.”
3. This brings up an important point: Why were the artifacts (9 from the internal validation and 35 from the external validation sets removed? This inappropriately inflates performance and is my main criticism of this paper. In my view, without these scans, the performance reported cannot be considered an estimate of real-world performance. Unless the algorithm has mechanisms to detect these problematic scans and report them separately. This was not discussed in the paper.

Author’s response: Thank you for your comment. We agree that it is an important aspect. Initially, we excluded the cases with severe artifacts because their images were heavily corrupted to be used for diagnosis. As suggested, we analyzed and reported the classification performance of the algorithm using entire datasets without any exclusion. However, the results with and without exclusion of cases did not differ significantly. Hence, clinical simulation tests were not performed again using the entire dataset. We respectfully hope that the reviewer understands our decision.

In the revised manuscript:

“In addition, to know the performance of the CN-StyleGAN under more real-world circumstances, it was also tested using the internal and external validation data without any exclusion. The AUC, sensitivity, specificity, and accuracy with 95% CIs in the internal validation test were 0.86 (0.81–0.89), 0.93 (0.86–0.93), 0.65 (0.61–0.69), and 0.69 (0.65–0.73), respectively, and those in the external validation test were 0.86 (0.83–0.89), 0.84 (0.78–0.89), 0.71 (0.69–0.73), and 0.72 (0.70–0.74), respectively. These results were not significantly different from the results obtained after excluding cases with metal artifacts.”

(Page 6, Line 100 – 106 in the Result)

4. Another criticism is that the algorithm is more of an anomaly detection than a critical finding identification system. This has to be stated as a limitation of the algorithm and not an issue of the validation set as stated in the manuscript “While our model did not achieve extremely accurate performance, this may be due to our validation dataset that included many anomalous cases but not requiring emergency intervention (for example, encephalomalacia, severe leukoaraiosis, or arachnoid cysts in the false positives).” I also disagree with the statement that false positives are not a problem for triage systems. They increase noise, decrease reliability of the system, and may impact trust. Also, regarding the following statement: “Moreover, the strength of our method, a triage tool that is explainable and generally applicable for various diseases, alleviates the need for achieving extremely accurate performance.” I agree that explainability is an extremely helpful feature and may expedite review, but I am not sure that the strength of the method (which again should be judged by the reader) should alleviate issues with performance.

Author’s response: We agree with the reviewer’s point of view. Therefore, we have revised the limitation accordingly. Please note that we have removed the term “explainable” or “explainability” in the revised manuscript according to the suggestion of the reviewer #1.

In the revised manuscript:

- “In addition, our algorithm is an anomaly detection model, rather than an identification model, for detecting critical findings. Therefore, our algorithm had problems in detecting anomalies of benign conditions and reported them as false positives (e.g., encephalomalacia, severe leukoaraiosis, and arachnoid cysts). These false positives can affect the diagnosis of

radiologists and have a negative effect on the radiology workflow. In addition, they are directly related to the reliability of the triage system. This concern has been reported in the previously published literature on triaging cases with artificial intelligence^{15,21}. To mitigate these problems, the anomaly detection model should be updated over time. However, unless this triage algorithm has perfect accuracy, the problems will not be solved. Therefore, the clinical relevance of our triage system for the outcomes should be carefully interpreted by measuring its net value. However, the retrospective study design has limitations for evaluating this issue. Therefore, further randomized controlled trials investigating the actual effect of these triage systems in clinical practices are warranted.”

(Page 9 – 10, Line 194 – 206 in the Discussion)

- “Moreover, our system visually highlighted the model’s prediction, which provided readers an independent review of the recommendation of the model, although the final decision should be made by the readers.”

(Page 8, Line 169 – 171 in the Discussion)

5. Another statement that seems inappropriate is the following: “However, in diagnosing most cases with emergency neurologic lesions, a typical image finding is pathognomonic by itself.” I do not think this is true. Imaging findings are rarely pathognomonic, they are typically associated with differential diagnosis, and not having prior imaging to compare makes this even more complicated.

Author’s response: We agree with the reviewers’ opinion on the diagnosis of emergency neurological lesions. Nevertheless, we hope that the reviewer understands that this is an unavoidable limitation. To evaluate the real-world performance of the algorithm, we tested the algorithm using the data of the internal and external validation datasets, which included consecutive patients who underwent emergency screening in the emergency department. In many cases of these settings, pathological diagnosis in many cases of these screening cohorts was not easy. For example, ischemic stroke is usually diagnosed based on clinical information (patient’s history, symptoms, and neurological examination) and radiological examination and is preferably treated with medical care, rather than invasive treatment. In the case of brain metastasis, the diagnosis is made based on the underlying primary cancer and radiological examination. Although the radiological diagnosis in some cases (e.g., intracranial hemorrhage) is conclusive, it is frequently made to transfer patients to another hospital due to the limited operating room resources. We have added these problems in the limitation section of the revised manuscript.

In addition, because of the aforementioned limitation of our study, we changed the diagnostic term from “brain tumor” to “brain mass-like lesions” as reported in a previous clinical study. The brain mass-like lesions are defined as volumetric space-occupying lesions (e.g., brain tumor, brain abscess, tumefactive demyelinating disease, and encephalitis) distinct from the normal-appearing brain parenchyma.

In the revised manuscript:

- “Third, this study referred to the clinical and radiological diagnoses rather than the histopathological diagnoses. To evaluate the real-world performance of the algorithm, we tested the algorithm in the emergency screening cohorts. In many cases of these cohorts, the pathological diagnosis was particularly difficult because of the preference for non-invasive diagnosis (e.g., ischemic stroke) and treatment, inoperable status of patients, or transfer of patients to another facility due to limited operable resources. Therefore, it might have been an unavoidable limitation of this study. Nevertheless, further studies assessing the accuracy of our triage system with the gold standard are warranted.”

(Page 10, Line 216 – 223 in the Discussion)

- “The diseases entities in the emergency cases were categorized into brain mass-like lesions, acute infarctions, intracranial hemorrhages, hydrocephalus, and other diseases. A brain mass-like lesion was defined as a volumetric space-occupying lesion (e.g., brain tumor, brain abscess, tumefactive demyelinating disease, and encephalitis) distinct from the normal-appearing brain parenchyma.”

(Page 13, Line 273 – 277 in the Method)

6. I also think that not considering relevant prior imaging studies as part of the decision making to determine whether a study is emergent or not is a limitation that needs to be clearly stated.

Author’s response: We thank the reviewer for this suggestion. The fact that we did not consider the findings of previous imaging studies in decision-making has been mentioned as a limitation in the revised manuscript.

In the revised manuscript:

“Second, our current system is based on a single brain CT scan without referring to prior image examinations and clinical information. This could cause mis-triage of some less urgent cases as a high prioritization case. For example, even if a previously diagnosed infarction has already been treated, it could be detected as an emergency case.”

(Page 10, Line 209 – 212 in the Discussion)

7. And finally, the authors should expand on the anomaly score system and provide examples. A detailed explanation of how the cutoff of the anomaly score was derived should be presented. Was this performed during tuning or validation?

Author’s response: Thank you for the suggestion. We have described the anomaly score system in detail in the Methods section and have provided examples in Supplementary Figure 5. In addition, we have explained the method of cutoff value derivation. The cutoff value was the maximum value of Youden’s index for ROC curve of the internal validation test. The same cutoff value was applied to both the internal and external validation tests. As this has been illustrated only in the Method section, we have made the additional statement in the main article to make it clear how the cutoff was derived.

In the revised manuscript and supplementary materials:

- “*Anomaly score.* The anomaly score was calculated as follows: first, reconstruction error for a slice x_i of a query scan was defined as $R(x_i) = \|M \odot (x_i - G(w, n))\|$, which is the binary masked density error between the query slice x and the synthetic slice $G(w, n)$. Second, this reconstruction error was normalized, slice-by-slice, based on the slice order, using the reconstruction error statistics of the mean R_μ and SD R_σ of normal brain CT images. A total of 1,000 scans were randomly selected from the training dataset for normal brain reconstruction error statistics. Finally, this normalized per-slice reconstruction error of 32 slices for the query scan was summed up to obtain the anomaly score: *Anomaly score* =

$$\sum_{i=1}^{32} \frac{R(x_i) - R_{\mu_i}}{R_{\sigma_i}}$$

(Page 15, Line 331 – 338 in the Method)

- **Supplementary Figure 5.** Inference method and anomaly score system of the CN-StyleGAN

The figure demonstrates the inference method and anomaly score system of the CN-StyleGAN. **a**, The inference method of the CN-StyleGAN. Given a query brain CT slice x , the latent vector w was initialized as $E(x)$ and the noise maps n were initialized from the unit normal distribution. After the latent vector optimization, the noise maps were optimized. Masked noise optimization was proposed for the noise map optimization. **b**, Derivation of the binary mask in the masked noise optimization process. The residual difference between a query image and reconstructed image is brain-extracted, median-filtered, and thresholded.

Moreover, the false positives in the binary mask were reduced because of the intersections of binary masks in the previous optimization steps. **c**, The calculation process of the anomaly score system. The reconstruction error of a slice was derived as binary masked density error between the query slice and the synthetic slice. This reconstruction error was normalized, slice-by-slice based on the slice order, using the reconstruction error statistics (mean and standard deviation of normal brain CT images) from the training dataset. Finally, this normalized per-slice reconstruction error of 32 slices for a query scan was summed up to determine the anomaly score.

- “ROC curve analysis in the internal validation test revealed the optimal anomaly score cutoff, which was derived from the maximum value of Youden's index.”

(Page 5, Line 87 – 89 in the Result.)

Reviewers' Comments:

Reviewer #1:

Remarks to the Author:

I thank the authors for their modifications and honest answers to my questions and comments.

There is one remaining issue, and I believe it is quite crucial. This is related to the recall bias. Authors mentioned that two radiologists read the same images two weeks apart. The time gap is placed to reduce the possible recall bias. At the same time, they report that using the algorithm WT and TAT of both emergency and non-emergency images decreased, with the latter showing a smaller decrease.

As discussed in the review process, reducing the time spent reading a block is supposed to be a zero-sum game. However, this is not the case here. The overall reading time of the block reduced. This raises a very critical question: is this the result of the recall bias?

Answering this question is not possible in the current setup as far as I can see, unless the same experiment is repeated on multiple blocks with blocks being randomly separated into those that will be read without and those with the help of the triage algorithm.

Given the current experimental setup, I do not see how the experimental setup truly can support the conclusion that the triage method reduces the WT and TAT.

Reviewer #2:

Remarks to the Author:

I believe the changes made by the authors considerably increased the quality of the manuscript. There are a few of the responses that still require some additional work.

Regarding the following statements: "Although the anomaly detection framework in the previous studies has attracted considerable attention, they have two limitations. First, the anomaly detection framework lacks external and clinical validation tests, hence, whether the model can generalize to real-world situations cannot be guaranteed. Second, the anomaly detection framework operates on low-resolution images. For example, a recent study used image patches with a resolution of 64×64 pixels¹¹, which is not applicable to original CT images having a resolution of 512×512 pixels." Would be more clear to state in the last two phrases that they relate to prior studies. The way it stands it is a bit confusing. Maybe this could be solved the following way: "Although anomaly detection frameworks reported on previous studies has attracted considerable attention, they have two limitations: 1) lack of external and clinical validation tests, hence, whether the model can generalize to real-world situations cannot be guaranteed; 2) operate on lower-resolution images (e.g. 64×64)¹¹ when compared to 512×512 pixels on CT."

I think selecting the score cutoff based on internal validation test (the last phase of internal validation) is biased and incorrect. This would have to be performed during early stages of algorithm tuning and selection. My recommendation is to re-run the analysis based on the correct approach. Luckily, there is an external validation set that suggests there will not be major changes in results, but I believe this needs to be corrected.

Regarding the following statement: "These results were not significantly different from the results obtained after excluding cases with metal artifacts." This implies statistical analysis. However, not even the results are not even presented. So I suggest this statement to be removed or fully

reported and analyzed.

The acronym WT needs to be re-introduced here: "The clinical efficacy of the model was analyzed using three radiological time metrics based on previous studies¹²⁻¹⁴: WT, radiology report turnaround time (TAT), and reading time (RT) for each case in each block."

How can the 18.0 s [14–27 s] vs. 18.0 s [14–27 s], $P < .001$ be statistically significant? "In the non-emergency and emergency groups, a significant difference was noted in the RTs between two reading sessions (22 s [IQR: 17–30 s] vs 17 s [IQR: 14–23 s], $P < .001$; 18.0 s [14–27 s] vs. 18.0 s [14–27 s], $P < .001$)."

The following new paragraph is very confusing and has a lot of point/counterpoints that go back and forth. Sometimes it is hard to understand the main message is and some of the phrases do not connect with the what as stated previously: "It is noteworthy that the proposed triage system can reprioritize emergency cases, leading to the reduction of WT and TAT. This result means that this system could be one of the strategies to improve workflow efficiency in neurological emergencies by reducing wait times for radiology reporting and acute care. However, there are a number of issues that need to be resolved before actual clinical application. Ironically, these time metrics decreased in non-emergency cases, although they were read later. Given the small reduction of RT in both the non-emergency and emergency groups, it is reasonable to presume that prioritization causes behavioral changes in radiologists, which was also observed in another study on a triage system¹⁶. It appears to be some type of automation bias^{19,20}, wherein readers over-rely on the software recommendation, which may then decrease the RT and subsequently affect WT and TAT. Nevertheless, compared with that in the emergencies, the degree of these reductions in WT and TAT was relatively more minor and thus may be clinically insignificant. Thus, we think this automation bias has minimal influence on the triage algorithm-driven positive effect in the present study. However, our results showed that this triage system has a risk of delayed reporting of false negatives, although this result was not statistically significant and the change in time metrics was relatively smaller in false-negative cases than in emergency cases. Therefore, given the limited resources in the emergency setting, it could be a zero-sum game because the gain in some patients is offset by the loss in other patients. In addition, our algorithm is an anomaly detection model, rather than an identification model, for detecting critical findings. Therefore, our algorithm had problems in detecting anomalies of benign conditions and reported them as false positives (e.g., encephalomalacia, severe leukoaraiosis, and arachnoid cysts). These false positives can affect the diagnosis of radiologists and have a negative effect on the radiology workflow. In addition, they are directly related to the reliability of the triage system. This concern has been reported in the previously published literature on triaging cases with artificial intelligence^{15,21}. To mitigate these problems, the anomaly detection model should be updated over time. However, unless this triage algorithm has perfect accuracy, the problems will not be solved. Therefore, the clinical relevance of our triage system for the outcomes should be carefully interpreted by measuring its net value. However, the retrospective study design has limitations for evaluating this issue. Therefore, further randomized controlled trials investigating the actual effect of these triage systems in clinical practices are warranted." I suggest this to be re-written.

Point-by-point responses to the comments from the reviewer

Manuscript ID: NCOMMS-21-32482B-Z

Title: Emergency triage of brain computed tomography via anomaly detection with a deep generative model

We appreciate your thoughtful review of our manuscript and hope that the reviewers find our revisions satisfactory. We have tried to address all queries.

As the reviewers suggested, we have re-performed the clinical simulation tests using a randomized crossover study design to mitigate a recall bias or learning effect. This process was lengthy due to the study design having a large number of cases and washout periods. For this delay, we ask for the understanding of the reviewers. The paper has been modified considering the additional analysis and the updated references.

Please note that the revised portions are highlighted in blue for your convenience in the revised manuscript and supplementary material.

Reviewer #1 (Remarks to the Author):

Q1) I thank the authors for their modifications and honest answers to my questions and comments.

There is one remaining issue, and I believe it is quite crucial. This is related to the recall bias. Authors mentioned that two radiologists read the same images two weeks apart. The time gap is placed to reduce the possible recall bias. At the same time, they report that using the algorithm WT and TAT of both emergency and non-emergency images decreased, with the latter showing a smaller decrease.

As discussed in the review process, reducing the time spent reading a block is supposed to be a zero-sum game. However, this is not the case here. The overall reading time of the block reduced. This raises a very critical question: is this the result of the recall bias?

Answering this question is not possible in the current setup as far as I can see, unless the same experiment is repeated on multiple blocks with blocks being randomly separated into those that will be read without and those with the help of the triage algorithm.

Given the current experimental setup, I do not see how the experimental setup truly can support the conclusion that the triage method reduces the WT and TAT.

Re) We thank the reviewer and agree with the valuable comment. To solve this problem, we have decided to run the clinical simulation study again. To mitigate the recall bias and the learning effect, we

have adopted a randomized crossover study design with a washout period of two weeks with reference to an existing study [1]. A total of 1795 cases from the consecutive external validation dataset were randomized to two groups (group A [898 cases] and group B [897 cases]). In the first session, each reader assessed group A without the deep learning-based triage and group B with the deep learning-based triage. In the second session, each reader assessed group A with the deep learning-based triage and group B without the deep learning-based triage. The first and second sessions were separated by at least two weeks, and the reading order of the blocks was randomized for each reading session to reduce recall bias. In addition, to mitigate the effect of the reader's expertise, we recruited a new reader having the same experience level who was not involved in the previous experiment. According to the request by reviewer 2, this simulation test included the complete validation datasets without any exclusion.

Fig. 5a: Randomized crossover study design. ADA, anomaly detection algorithm

In the emergency group, median WT was significantly 294 s shorter in the post-ADA group (70.5 s [IQR 168]) in comparison with the pre-ADA group (422.5 [IQR 299]) ($p < 0.001$). Median TAT was significantly 297.5 s faster in the post-ADA group (88.5 s [IQR 179]) in comparison with the pre-ADA group (445.0 s [IQR 298]) ($p < 0.001$). There was no significant difference in RT between pre-ADA and post-ADA (29.0 s [IQR 12.5] vs. 30.0 s [IQR 11.0], $p = 0.38$). In the non-emergency group, there was a significant delay in the WT and TAT when the ADA was implemented. The RT was significantly 1.5 s shorter in the post-ADA group (31.00 s [11.5]) in comparison with the pre-ADA group (28.00 [11.5]) ($p < 0.001$). However, the increase of WT and TAT in the non-emergency group (79.3 s [IQR 197.9] and 72.8 s [IQR 202.3]) was significantly smaller than the decrease in WT and TAT in the emergency

group (-294.0 s [IQR 352] and -297.5 s [IQR 347]) ($p < 0.001$). It is likely due to the small percentage of emergency cases in our cohort along with shorter RT after ADA implementation in non-emergency cases. Although the emergency cases lead to radiology workflow delay in the control group, the shorter RT in the relatively larger control group seems to offset the effects. Given our study design of clinical simulation test, the shorter RT in the non-emergency cases may be due to the change in the radiologists' confidence or behavior for image interpretation in the normal brain CTs predicted by ADA rather than recall bias or learning effect. However, this issue needs further studies. We have added these contents in the revised manuscript.

Reference

[1] Sung, J., Park, S., Lee, S. M., Bae, W., Park, B., Jung, E., Seo, J. B., & Jung, K. H. (2021). Added Value of Deep Learning-based Detection System for Multiple Major Findings on Chest Radiographs: A Randomized Crossover Study. *Radiology*, 299(2), 450–459. <https://doi.org/10.1148/radiol.2021202818>

Reviewer #2 (Remarks to the Author):

I believe the changes made by the authors considerably increased the quality of the manuscript. There are a few of the responses that still require some additional work.

Q1) Regarding the following statements: "Although the anomaly detection framework in the previous studies has attracted considerable attention, they have two limitations. First, the anomaly detection framework lacks external and clinical validation tests, hence, whether the model can generalize to real-world situations cannot be guaranteed. Second, the anomaly detection framework operates on low-resolution images. For example, a recent study used image patches with a resolution of 64×64 pixels¹¹, which is not applicable to original CT images having a resolution of 512×512 pixels." Would be more clear to state in the last two phrases that they relate to prior studies. The way it stands it is a bit confusing. Maybe this could be solved the following way: "Although anomaly detection frameworks reported on previous studies has attracted considerable attention, they have two limitations: 1) lack of external and clinical validation tests, hence, whether the model can generalize to real-world situations cannot be guaranteed; 2) operate on lower-resolution images (e.g. 64×64)¹¹ when compared to 512×512 pixels on CT."

Re) We appreciate the reviewer's kind comment. We have revised our manuscript as follows, and have updated the reference list with recently published papers. Therefore, we have removed the context for

the resolution considering the recently published papers, and we have updated the context with reference to recently published papers.

In the revised manuscript:

“Although previous studies using this framework have attracted considerable attention, they have two limitations: 1) lack of external and clinical validation tests, hence, whether the model can generalize to real-world situations cannot be guaranteed; 2) no clinical utility test of the deep generative models.”
(Page 3, Line 36 – 39)

“Our results are supported by previous relative studies in terms of acceptable performance by an anomaly detection model and good generalizability. Han et al. reported on a GAN-based anomaly detection model having an AUC of 0.727-0.894 in detecting Alzheimer’s diseases, and having an AUC of 0.921 in detecting brain metastases from MRI. Choi et al. reported on a deep learning model trained only by normal brain to identify brain abnormalities (AUC of 0.74) in brain positron emission tomography-CT (PET-CT). Fujioka et al. proposed anomaly detection using GAN with an AUC of 0.936 in distinguishing normal tissue from benign and malignant masses in breast ultrasound imaging. These prior studies are valuable in that they demonstrated the capability of anomaly detection models in various medical images. However, the previous studies lacked external clinical validation tests, hence, whether the model could generalize to real-world situations cannot be guaranteed. Therefore, further evidence with real-world data is warranted. Our study serves this purpose.” (Page 8, Line 151 – 162)

Q2) I think selecting the score cutoff based on internal validation test (the last phase of internal validation) is biased and incorrect. This would have to be performed during early stages of algorithm tuning and selection. My recommendation is to re-run the analysis based on the correct approach. Luckily, there is an external validation set that suggests there will not be major changes in results, but I believe this needs to be corrected.

Re) We agree with the reviewer’s comment and have corrected the analysis and results as recommended. In the revised paper, the internal dataset was randomly divided into two halves for the tuning and internal validation tests. Subsequently, all the cutoff values were selected using the tuning dataset.

In the revised manuscript:

“The internal dataset included brain CT scans of 544 individuals (mean age \pm SD: 58.6 ± 17.8 years; women: 280 [51.5%]) who had visited the ED of the internal institution for 1 month. Following that, the internal dataset was randomly divided into two parts: the tuning dataset and the internal validation

dataset. The external validation dataset included brain CT scans of 1,795 consecutive individuals (mean age \pm SD: 60.3 \pm 19.3 years; female: 875 [48.7%]) who had visited the ED of an external institution for 5 months.” (Page 4, Line 59 – 64)

Q3) Regarding the following statement: "These results were not significantly different from the results obtained after excluding cases with metal artifacts." This implies statistical analysis. However, not even the results are not even presented. So I suggest this statement to be removed or fully reported and analyzed.

Re) We agree with the reviewer’s comment that the statement should be removed or fully reported and analyzed. In the revised paper, all experiments and results were performed and analyzed with no exclusions from the validation datasets. Thus, the statement was removed.

Q4) The acronym WT needs to be re-introduced here: "The clinical efficacy of the model was analyzed using three radiological time metrics based on previous studies¹²⁻¹⁴: WT, radiology report turnaround time (TAT), and reading time (RT) for each case in each block."

Re) We thank the reviewer for the comment. We have re-introduced the acronym WT and the definition of time metrics briefly, as follows:

In the revised manuscript:

“The clinical efficacy of the ADA was analyzed according to three radiological time metrics based on previous studies: wait time (WT; the time required to open a CT for image review from the beginning of one block), radiology report turnaround time (TAT; the time required to report a critical CT finding from the beginning of one block), and reading time (RT; the time between opening and closing a CT) for each case in each block.” (Page 6, Line 118 – Page7, Line 112)

Q5) How can the 18.0 s [14–27 s] vs. 18.0 s [14–27 s], $P < .001$ be statistically significant? "In the non-emergency and emergency groups, a significant difference was noted in the RTs between two reading sessions (22 s [IQR: 17–30 s] vs 17 s [IQR: 14–23 s], $P < .001$; 18.0 s [14–27 s] vs. 18.0 s [14–27 s], $P < .001$)."

Re) We appreciate the reviewer’s valuable comment. Before we answer, we would like to inform the reviewer that our clinical simulation test was performed again according to reviewer #1’s comment. This clinical simulation test included all the validation data, without any exclusion, to reflect the real-

world data. In addition, to alleviate the possibility of statistical errors, the statistician analyzed all outcomes in the re-performed test. We have revised our manuscript to reflect the study design and results. We hope the reviewer understand this explanation.

For the reviewer's question, we offer the response below, based on the statistician's advisement:

As to your comment about our previous result, we agree that the result is surprising, as a significant difference was noted, whereas the median of the reading times between two reading sessions was identical. However, according to the statistician's advisement, this is a somewhat common finding. In fact, the key to this issue is that the Mann-Whitney and the equivalent Wilcoxon test (also known as the Mann-Whitney-Wilcoxon test) are rank sum tests and not median tests [1]. There exist many examples of identical medians with a very small Wilcoxon p-value.

The two papers listed below demonstrate this well:

[1] <https://stats.oarc.ucla.edu/other/mult-pkg/faq/general/faq-why-is-the-mann-whitney-significant-when-the-medians-are-equal/>

[2] <https://journals.sagepub.com/doi/pdf/10.1177/1536867X1201200202>

At the beginning of Chapter 4 of this paper [2], "The Mann-Whitney test is a test for equality of medians only under the very strong assumption that both of the two distributions are symmetrical about their respective medians or, in the case of asymmetric distributions, that the distributions are of the same shape but differ in location. Thus the common belief that the test compares medians is true only under some implausible circumstances." In other words, there are some cases where the Wilcoxon rank sum test (=Mann-Whitney test) is done to determine if the medians are identical, but in that case, it is only done under implausible circumstances.

Q6) The following new paragraph is very confusing and has a lot of point/counterpoints that go back and forth. Sometimes it is hard to understand the main message is and some of the phrases do not connect with the what as stated previously: "It is noteworthy that the proposed triage system can reprioritize emergency cases, leading to the reduction of WT and TAT. This result means that this system could be one of the strategies to improve workflow efficiency in neurological emergencies by reducing wait times for radiology reporting and acute care. However, there are a number of issues that need to be resolved before actual clinical application. Ironically, these time metrics decreased in non-emergency cases, although they were read later. Given the small reduction of RT in both the non-emergency and emergency groups, it is reasonable to presume that prioritization causes behavioral changes in radiologists, which was also observed in another study on a triage system¹⁶. It appears to be some type

of automation bias^{19,20}, wherein readers over-rely on the software recommendation, which may then decrease the RT and subsequently affect WT and TAT. Nevertheless, compared with that in the emergencies, the degree of these reductions in WT and TAT was relatively more minor and thus may be clinically insignificant. Thus, we think this automation bias has minimal influence on the triage algorithm-driven positive effect in the present study. However, our results showed that this triage system has a risk of delayed reporting of false negatives, although this result was not statistically significant and the change in time metrics was relatively smaller in false-negative cases than in emergency cases. Therefore, given the limited resources in the emergency setting, it could be a zero-sum game because the gain in some patients is offset by the loss in other patients. In addition, our algorithm is an anomaly detection model, rather than an identification model, for detecting critical findings. Therefore, our algorithm had problems in detecting anomalies of benign conditions and reported them as false positives (e.g., encephalomalacia, severe leukoaraiosis, and arachnoid cysts). These false positives can affect the diagnosis of radiologists and have a negative effect on the radiology workflow. In addition, they are directly related to the reliability of the triage system. This concern has been reported in the previously published literature on triaging cases with artificial intelligence^{15,21}. To mitigate these problems, the anomaly detection model should be updated over time. However, unless this triage algorithm has perfect accuracy, the problems will not be solved. Therefore, the clinical relevance of our triage system for the outcomes should be carefully interpreted by measuring its net value. However, the retrospective study design has limitations for evaluating this issue. Therefore, further randomized controlled trials investigating the actual effect of these triage systems in clinical practices are warranted." I suggest this to be re-written.

Re) We apologize for any ambiguity in our writing. According to the reviewer's comment, we have revised our manuscript as follows:

In the revised manuscript:

"The other critical point of our study is that our research demonstrated the feasibility of our ADA as a triage system of brain CTs in the ED. Our study revealed that the ADA implementation significantly reduced the WT and TAT in emergency cases. Our result is comparable to those of previous studies regarding the clinical feasibility of patient triage by supervised anomaly detection models. Titano et al.¹⁸ reported that their supervised model potentially raises the alarm 150 times faster than humans for urgent cases in brain CTs. Wood et al.¹⁹ demonstrated that the supervised anomaly detection model significantly reduced the mean reporting time for abnormal MRI examinations from 28 days to 14 days and from 9 days to 5 days for two hospital networks. Notably, in the detailed subgroup analysis of our study, the ADA implementation led to a significant reduction in the WT and TAT in the more urgent

(immediate) group than the urgent group. This is because ADA-based classification is based on anomaly scores. A higher anomaly score for a limited intracranial space likely reflects a correspondent urgency in emergency brain CTs. Unlike our expectation, the increase of WT and TAT in the non-emergency group was significantly smaller than the decrease in WT and TAT in the emergency group. It is likely due to the small percentage of emergency cases and shorter RT following ADA implementation in non-emergency cases. Although the emergency cases lead to radiology workflow delay in the control group, the faster RT in the relatively larger control group seems to offset the effects. Given our study design of clinical simulation test, the shorter RT in the non-emergency cases may be due to the change in the radiologists' confidence or behavior for image interpretation in the normal brain CTs predicted by ADA rather than recall bias or learning effect. However, this issue needs further studies.

The unresolved problem for anomaly detection models is the relatively high false positive and false negative rates. In the randomized controlled study conducted by Titano et al.¹⁸, their supervised model for the triage of urgent brain CTs could alert physicians in 50% of critical cases, with a 21% false-alarm rate. Our model had a high false negative rate (22.3%) and false positive rate (19.1%). In our clinical simulation test, the ADA implementation caused a significant delay in median WT and TAT in the false negatives in comparison with the control group. Therefore, the triage system with the anomaly detection model posed the risk of undermining the timely management of patients with critical CT findings. For false positives, the false alarm can reduce physicians' faith in the model and negatively affect emergency patients who need fast treatment. Although these problems could be solved using technical advances, this will be an ongoing issue unless the triage algorithm achieves perfect accuracy. Therefore, it is important that the interpreting radiologists understand the optimization strategy and are prepared to deal with false positives or negatives.

This study has several limitations. First, our current system relies on a single brain CT scan and does not refer to prior image examinations or clinical information. This could result in the mis-triage of some less urgent cases as high prioritization cases. For example, even if a previously diagnosed infarction has already been treated, it could be detected as an emergency case. Besides, the anomaly cases of benign conditions (e.g., arachnoid cyst or encephalomalacia with old infarction) can also be incorrectly classified into emergency conditions. In addition, brain shrinkage is a normal part of the aging process but can indicate early-onset neurodegenerative diseases in younger patients. Therefore, generating the closest normal brain images without age information is challenging. Age information could be the prerequisite for correct classification in our anomaly detection model. These problems can be mitigated by training the model on benign conditions and incorporating meta-information regarding the factors that affect clinical diagnosis. Third, we used clinical and radiological diagnoses as reference standards. However, many neurological ED cases (e.g., small traumatic intracranial hemorrhage, minor stroke, or transient ischemic attack) do not require surgical treatment or aggressive intervention because of their low risk of rapid exacerbation. Therefore, it could be an unavoidable limitation in the emergency

screening cohort study. Nevertheless, further studies using the gold standard are warranted to know the accurate performance of the model. Fourth, this study did not reflect the complexity of clinical practice. Multiple factors can influence the results of a clinical simulation test, including case difficulty, the queue size of the CT scan, readers' expertise level, image-processing time, patient acuity, and interruption by other examinations. Therefore, our results may vary with these factors. To address this issue, multicentered and prospective validation studies are warranted.” (Page 8, Line 163 – Page10, Line 213)

Reviewers' Comments:

Reviewer #1:

Remarks to the Author:

I thank the authors once again for their willingness to adapt the study. I am convinced with the new experimental setup. All my questions have been answered.

Reviewer #2:

Remarks to the Author:

I think the revisions are satisfactory from a content standpoint. I do believe however that the manuscript would benefit from a review for language and grammar. An example is the phrase "Our results are supported by previous relative studies in terms of acceptable performance by an anomaly detection model and good generalizability". The authors probably meant to use the word related and not relative. The construction of several phrases could also be improved.

Point-by-point responses to the comments from the reviewer

Manuscript ID: NCOMMS-21-32482C

Title: Emergency triage of brain computed tomography via anomaly detection with a deep generative model

We thank you for your positive consideration of our paper. The manuscript and figures have been modified according to the reviewers' comments and the guidance proposed by *Nature Communications*.

REVIEWERS' COMMENTS

Reviewer #1 (Remarks to the Author):

Q1) I thank the authors once again for their willingness to adapt the study. I am convinced with the new experimental setup. All my questions have been answered.

Re) We thank you for providing critical inputs that proved to be helpful in revising and improving this manuscript.

Reviewer #2 (Remarks to the Author):

Q1) I think the revisions are satisfactory from a content standpoint. I do believe however that the manuscript would benefit from a review for language and grammar. An example is the phrase "Our results are supported by previous relative studies in terms of acceptable performance by an anomaly detection model and good generalizability". The authors probably meant to use the word related and not relative. The construction of several phrases could also be improved.

Re) As the reviewer suggested, a professional English editor has reviewed our manuscript. We appreciate your thoughtful review of our manuscript and hope that the reviewer finds our revisions satisfactory.